New phiomorph rodents from the latest Eocene of Egypt, and the impact of Bayesian “clock”-based phylogenetic methods on estimates of basal hystricognath relationships and biochronology

Sallam Hesham M. 1 2 sallam@mans.edu.eg
Seiffert Erik R. 3 erik.seiffert@gmail.com erik.seiffert@stonybrook.edu
1 Vertebrate Paleontology, Department of Geology, Faculty of Science, Mansoura University , Mansoura , Egypt
2 Department of Evolutionary Anthropology, Duke University , Durham, NC , United States
3 Department of Anatomical Sciences, Stony Brook University , Stony Brook, NY, United States , USA
Druzinsky Robert
Electronic publication date: 2016 Mar 1
Publication date: 2016
Volume: 4
Electronic Location ID: e1717
Received 2015 Dec 1; Accepted 2016 Jan 31
Copyright: ©2016 Sallam and Seiffert
Copyright year: 2016
Copyright holder: Sallam and Seiffert
License: This is an open access article distributed under the terms of the Creative Commons Attribution License, which permits unrestricted use, distribution, reproduction and adaptation in any medium and for any purpose provided that it is properly attributed. For attribution, the original author(s), title, publication source (PeerJ) and either DOI or URL of the article must be cited.
License URL: https://creativecommons.org/licenses/by/4.0/

Keywords: Phiomorpha, Deciduous teeth, Oligocene, Paleogene, Africa, Bayesian phylogenetics, Hystricognathi

Funding: US National Science Foundation (NSF) grant BCS-0416164 BCS-0819186 BCS-1231288 The Leakey Foundation This research was funded by US National Science Foundation (NSF) grant BCS-0416164 to Erik R. Seiffert and Elwyn L. Simons, NSF BCS-0819186 to Erik R. Seiffert, NSF BCS-1231288 to Erik R. Seiffert, John G. Fleagle, Gregg F. Gunnell, and Doug M. Boyer, as well as grants from The Leakey Foundation to E.R.S. and H.M.S. The funders had no role in study design, data collection and analysis, decision to publish, or preparation of the manuscript.

==============================
The Fayum Depression of Egypt has yielded fossils of hystricognathous rodents from multiple Eocene and Oligocene horizons that range in age from ∼37 to ∼30 Ma and document several phases in the early evolution of crown Hystricognathi and one of its major subclades, Phiomorpha. Here we describe two new genera and species of basal phiomorphs, Birkamys korai and Mubhammys vadumensis, based on rostra and maxillary and mandibular remains from the terminal Eocene (∼34 Ma) Fayum Locality 41 (L-41). Birkamys is the smallest known Paleogene hystricognath, has very simple molars, and, like derived Oligocene-to-Recent phiomorphs (but unlike contemporaneous and older taxa) apparently retained dP4∕4 late into life, with no evidence for P4∕4 eruption or formation. Mubhammys is very similar in dental morphology to Birkamys, and also shows no evidence for P4∕4 formation or eruption, but is considerably larger. Though parsimony analysis with all characters equally weighted places Birkamys and Mubhammys as sister taxa of extant Thryonomys to the exclusion of much younger relatives of that genus, all other methods (standard Bayesian inference, Bayesian “tip-dating,” and parsimony analysis with scaled transitions between “fixed” and polymorphic states) place these species in more basal positions within Hystricognathi, as sister taxa of Oligocene-to-Recent phiomorphs. We also employ tip-dating as a means for estimating the ages of early hystricognath-bearing localities, many of which are not well-constrained by geological, geochronological, or biostratigraphic evidence. By simultaneously taking into account phylogeny, evolutionary rates, and uniform priors that appropriately encompass the range of possible ages for fossil localities, dating of tips in this Bayesian framework allows paleontologists to move beyond vague and assumption-laden “stage of evolution” arguments in biochronology to provide relatively rigorous age assessments of poorly-constrained faunas. This approach should become increasingly robust as estimates are combined from multiple independent analyses of distantly related clades, and is broadly applicable across the tree of life; as such it is deserving of paleontologists’ close attention. Notably, in the example provided here, hystricognathous rodents from Libya and Namibia that are controversially considered to be of middle Eocene age are instead estimated to be of late Eocene and late Oligocene age, respectively. Finally, we reconstruct the evolution of first lower molar size among Paleogene African hystricognaths using a Bayesian approach; the results of this analysis reconstruct a rapid latest Eocene dwarfing event along the lineage leading to Birkamys.

Introduction

The rodent clade Hystricognathi first appeared in the Eocene, and is now represented by three major groups with extant members—Hystricidae (Old World porcupines), Caviomorpha (New World hystricognaths), and Phiomorpha (African cane, dassie, and mole rats) (Singleton, Dickman & Stoddart, 2006). The largest DNA datasets currently available place Hystricidae as the sister group of a Caviomorpha-Phiomorpha clade (e.g., Huchon et al., 2007; Meredith et al., 2011; Patterson & Upham, 2014). Despite their modern distribution, being restricted almost entirely to southern continents, phylogenetic evidence provided by later Paleogene Asian “baluchimyines” suggests that the stem lineage of Hystricognathi probably arose in Asia (Marivaux et al., 2002; Marivaux, Vianey-Liaud & Jaeger, 2004; Sallam et al., 2009), though no members of the group are definitively known from that continent before the Eocene-Oligocene boundary (∼34 Ma: Marivaux et al., 2000; De Bruijn et al., 2003). In contrast, recent paleontological work in Tunisia and Peru has revealed that hystricognaths were present in Africa by ∼39.5 Ma (Marivaux et al., 2014), and that caviomorphs were present in South America by ∼41 Ma (Antoine et al., 2012).1 The latter discovery is critically important for establishing that the common ancestor of Caviomorpha and Phiomorpha must be even older than 41 Ma, and that stem members of Phiomorpha were already diversifying at least four million years prior to the deposition of the earliest well-sampled hystricognathous rodent fauna from Africa, the ∼37 Ma Birket Qarun Locality 2 in the Fayum Depression of Egypt (Sallam et al., 2009).

A recent study of molecular divergence estimates that took into account much of this new fossil evidence (Patterson & Upham, 2014) placed the caviomorph-phiomorph split at ∼42 Ma, the divergence of that clade from Hystricidae at ∼44.9 Ma, and the origin of crown Phiomorpha at ∼36.3 Ma. These estimates would suggest that the origin and initial diversification of crown Hystricognathi is not yet documented in the fossil record of any landmass, but that the origin of crown Phiomorpha should have occurred very close in time to the deposition of Locality BQ-2. Despite this, species that are known from ∼39 to ∼37 Ma African sites (Djebel el Kébar in Tunisia Marivaux et al., 2014, Bir el Ater in Algeria Jaeger, Denys & Coiffait, 1985, and BQ-2 in Egypt Sallam et al., 2009)—i.e., from a time period that would, given Patterson & Upham’s divergence estimates, postdate the caviomorph-phiomorph split by 3–5 Ma, and the origin of crown Hystricognathi by 6–8 Ma—have not been placed as stem phiomorphs in previous phylogenetic analyses (Sallam et al., 2009; Sallam, Seiffert & Simons, 2011; Sallam, Seiffert & Simons, 2012; Coster et al., 2010; Antoine et al., 2012), but instead are consistently placed as stem hystricognaths, or as stem members of the Caviomorpha-Phiomorpha clade. If these results are correct, then stem phiomorphs simply have not yet been sampled in the middle Eocene and early late Eocene sites of northern Africa.

One possible explanation for this pattern is that early phiomorphs have not yet been sampled due to a geographic bias, because all of the key sites documenting early hystricognath evolution in Africa are from the northernmost part of the continent. A reasonable alternative hypothesis, given the surprising discovery of ∼41 Ma caviomorphs and the poor early African record of this group, is that phylogenetic signal has been obscured by homoplasy between basal caviomorphs and more advanced stem phiomorphs, and some or all of the earliest African hystricognaths are actually basal stem phiomorphs that retain primitive morphology similar to that of the caviomorph-phiomorph ancestor. The possibility of early homoplasy between caviomorphs, phiomorphs, and the Asian “baluchimyine” radiation must be seriously entertained, because at present phylogenetic analyses of basal hystricognaths depend almost entirely on dental characters, many of which are known to have undergone remarkably rapid evolution in some early hystricognath lineages (notably Gaudeamuridae; Sallam, Seiffert & Simons, 2011). Compounding this problem is the fact that any phylogenetic arrangement of basal hystricognaths implies middle Eocene colonizations of large continents, which might have spurred rapid early diversification (and potentially rapid morphological change) associated with filling of open niche space (particularly in the case of South America, which lacked rodent competitors entirely).

Here we describe two new phiomorph genera and species from the latest Eocene Quarry L-41, in the Fayum area of northern Egypt (Fig. 1), that are the oldest to show suppression of P4∕4, one of the key dental synapomorphies of crown Phiomorpha. We include these and other basal African, Asian, and South American hystricognaths in a series of parsimony and Bayesian phylogenetic analyses, including Bayesian “tip-dating” analyses (Beck & Lee, 2014; Close et al., 2015; Dembo et al., 2015; Ronquist et al., 2012a) that are able to take into account information about the ages of fossil taxa, rates of morphological evolution, and models of speciation and extinction, and as such are potentially ideally suited to test relationships given the challenging circumstances presented by basal hystricognaths.

Figure 1 Major fossil localities in the Fayum Depression of northern Egypt.

(A) stratigraphic positions and age estimates for major mammal-bearing fossil localities in the Fayum succession, and approximate position of Eocene-Oligocene boundary (EOB), following Seiffert (2006). (B) map of northern Egypt showing common landmarks and localities near the Fayum Depression. (C) map of the Fayum area showing the approximate position of Locality 41 (L-41), which occurs near the middle of the section approximately midway between the oldest rodent-bearing fossil locality (BQ-2) and the youngest rodent-bearing fossil localities (I& M). Modified after Bown & Kraus (1988).

Aside from the tip-dating method’s obvious utility for phylogenetic reconstruction, we note that for species whose temporal ranges are poorly constrained by geological data, tip-dating takes into account both phylogenetic position and rate of morphological evolution to provide age estimates for those species, and this information provides a relatively rigorous testable hypothesis for the ordering of hystricognath-bearing faunas of Eocene and Oligocene age in Africa and Asia (i.e., DT1 and DT2 (Dur at-Talah, Libya), Lokone in Kenya, Silica North in Namibia, Paali Nala C2 and contemporaneous sites in the lower part of the Chitarwata Formation, Pakistan) that was not previously possible.

Material and Methods

Taxonomy

The electronic version of this article in Portable Document Format (PDF) will represent a published work according to the International Commission on Zoological Nomenclature (ICZN), and hence the new names contained in the electronic version are effectively published under that Code from the electronic edition alone. This published work and the nomenclatural acts it contains have been registered in ZooBank, the online registration system for the ICZN. The ZooBank LSIDs (Life Science Identifiers) can be resolved and the associated information viewed through any standard web browser by appending the LSID to the prefix http://zoobank.org/. The LSID for this publication is: urn:lsid:zoobank.org:pub:9DB0476B-E752-4EA1-8745-8C92E429C65B. The online version of this work is archived and available from the following digital repositories: PeerJ, PubMed Central and CLOCKSS.

Figure 2 Dental terminology, based on Wood & Wilson (1936) and Marivaux et al. (2014).

Terminology, measurements, and CT-scanning

Dental terminology follows Marivaux et al. (2014) (Fig. 2). Teeth are referred to as I, P, and M (for incisors, premolars, and molars, respectively), with upper and lower teeth designated by superscript and subscript numbers (respectively) for locus (e.g., the second lower molar is referred to as “M2”). Dental measurements were taken with a micrometer mounted in the lens of a Meiji binocular microscope. Specimens were scanned using a Nikon XT H 225 ST micro-CT scanner housed at at Duke University’s Shared Materials Instrumentation Facility and three-dimensional reconstructions were rendered in Avizo v. 8. Digital surface models (in Stanford “ply” format) of all specimens described here are available for viewing and direct download at www.morphosource.org.

Some of the Fayum rodent species described by Wood (1968) were revised by Holroyd (1994) as a part of her Ph.D. dissertation; taxonomic names that she considered to be invalid or incorrect are placed in quotation marks pending formal revision. Fossils are housed at the Egyptian Geological Museum (CGM) and Duke Lemur Center Division of Fossil Primates (DPC); a collection of casts is also housed in the Mansoura University Vertebrate Paleontology Center (MUVP) cast collection.

Identification of deciduous versus permanent premolars in early hystricognaths

Derived Oligocene-to-Recent phiomorphs have long been interpreted as having suppressed the eruption of permanent premolars, so that only dP4 and dP4 are present throughout life (e.g., Wood, 1968). The recent discoveries of middle Eocene “Protophiomys” tunisiensis (Marivaux et al., 2014), early late Eocene Protophiomys aegyptensis and Waslamys attiai (Sallam et al., 2009), late Eocene “Protophiomys” durattalahensis and “Protophiomys” aff. durattalahensis (Jaeger et al., 2010a), and latest Eocene Acritophiomys bowni (Sallam, Seiffert & Simons, 2012) and Gaudeamus aslius (Sallam, Seiffert & Simons, 2011), all of which retain the primitive pattern of replacing dP4∕4, have helped to clarify differences in the morphology of deciduous and permanent premolars in early hystricognaths and basal phiomorphs.

The P4s of all of these species are roughly ovoid and are either as long as they are broad, or are mesiodistally shorter than they are buccolingually broad; the primary lingual cusp is a centrally placed protocone, which is formed by a broad U-shaped crest that is isolated from buccal structures in the highly specialized genus Gaudeamus, but is connected to the protoloph and metaloph in all other early African species (the latter condition presumably representing the primitive state within crown Hystricognathi) (see Fig. S1 for comparative images). The P4s also lack enlarged hypocones (i.e., hypocones are much smaller than the protocones, or absent altogether). In contrast, the dP4s of Fayum hystricognaths bear occlusal surfaces that are very similar to those of the M1s, having mesially situated (rather than centrally placed) protocones, relatively large hypocones and distinct anterior arms of the hypocone, and capacious internal sinuses (or hypoflexuses), leading to a distinctly invaginated lingual margin rather than a strictly convex lingual margin. When compared with M1, the dP4 also tends to be relatively small, and have relatively weak lophs and cusps and a narrow lingual portion.

The P4s of early African hystricognaths are also roughly equal in length and width, have posterolophids that extend lingually all the way to the entoconid (rather than terminating at the point of the hypoconulid, or having only a very weak crest emanating lingually from the hypoconulid), and typically have a complete crest (metalophulid I) that connects the protoconid and metaconid across the mesial border of the tooth. In contrast, the dP4s of early African hystricognaths are mesiodistally elongate, have metaconids and protoconids that are closely approximated, have well-developed hypolophids that connect the entoconid to the hypoconid (or the anterior arm of the hypoconid), and have relatively large metaflexids (i.e., basins distal to the hypolophid). Some species also have crests on dP4 that course mesially from the protoconid. When taken in combination, these features have consistently allowed deciduous precursors and permanent replacement teeth to be identified in early African hystricognaths (Jaeger et al., 2010a; Jaeger et al., 2010b; Marivaux et al., 2012; Marivaux et al., 2014; Sallam et al., 2009; Sallam, Seiffert & Simons, 2011; Sallam, Seiffert & Simons, 2012), and we use the same criteria to identify the antemolar teeth of species described here as deciduous, rather than permanent.

Phylogenetic analysis

Matrix

The matrix employed here is that of Sallam, Seiffert & Simons (2012), which was built first on the original matrix of Marivaux, Vianey-Liaud & Jaeger (2004), and was then modified by Sallam et al. (2009), Sallam, Seiffert & Simons (2011) and Sallam, Seiffert & Simons (2012). The matrix contains 118 characters, mostly from the dentition, of which 77 were treated as ordered in all analyses; 97 of the characters are parsimony informative. Polymorphisms were assigned their own states that were situated between otherwise adjacent “fixed” states in ordered characters. Three additional early African species were added: “Protophiomys” tunisiensis from the late middle Eocene (Bartonian) of Djebel el Kébar, Tunisia (Marivaux et al., 2014); Turkanamys hexalophus, from the Oligocene Lokone Hill sites in the Turkana Basin, northern Kenya (Marivaux et al., 2012); and Prepomonomys bogenfelsi, from the Silica North site in the Sperrgebiet area of Namibia (Pickford et al., 2008), which is of contentious age, either Bartonian (Pickford et al., 2014) or significantly younger (Coster et al., 2012a; Marivaux et al., 2014). In all analyses, the early middle Eocene “chapattimyid” Birbalomys was designated as the outgroup.

Parsimony analyses (see Dataset S1) were run in PAUP 4.0b10 (Swofford, 2002) using the heuristic search algorithm, random addition sequence, and tree bisection-and-reconnection branch swapping across 10,000 replicates. Two weighting schemes were employed—one in which transitions between “fixed” and intermediate polymorphic states in ordered morphoclines were equal to a single step, and another in which those transitions were scaled to a half-step, so that transitions between “fixed” states were equal to a single step (as in Sallam, Seiffert & Simons, 2012). Bootstrap support was also calculated in PAUP, based on 1,000 pseudoreplicates.

Bayesian phylogenetic analyses (see Dataset S2) were run in MrBayes 3.2.5 (Ronquist et al., 2012a; Ronquist et al., 2012b). The Mk model for morphological data (Lewis, 2001) was used, coding was set to “variable,” and gamma-distributed rate variation across characters was assumed. Markov Chain Monte Carlo (MCMC) chains were run for 25 million generations, with two independent runs, each with one cold chain and three heated chains (temp = 0.02), sampling every 1,000 generations. The first 25% of the resulting 25,000 samples were discarded as the “burn-in” period, and the remaining trees were summarized using an “allcompat” (majority-rule plus compatible groups) consensus tree. Convergence was assessed by checking both effective sample sizes and the average standard deviation of split frequencies in the final generation.

Bayesian “tip-dating” analyses (see Dataset S3) were also run in MrBayes 3.2.5 (Ronquist et al., 2012a; Ronquist et al., 2012b). We employed the IGR (independent gamma rates) relaxed clock model and the fossilized birth–death prior on branch lengths, with “samplestrat” set to “fossiltip” (indicating that tips left no descendants). We ran several analyses with various perturbations of the priors “speciationpr” (the prior on the net speciation rate), “igrvarpr” (the prior on the variance of the gamma distribution from which branch lengths are drawn), and “clockratepr” (the prior on the base substitution rate, measured in number of changes per site per Ma) (Huelsenbeck, Ronquist & Teslenko, 2015), all of which yielded remarkably similar “allcompat” topologies, divergence dates, tip dates, and support values—however many of these analyses did not show adequate evidence for convergence across all parameters, as judged by effective sample sizes and potential scale reduction factors. Ultimately the combination of priors that yielded the strongest evidence for convergence across all parameters was speciationpr = exp(50), clockratepr = normal(0.25, 0.05), and igrvarpr = exp(3), with flat beta priors on fossilizationpr and extinctionpr, and we present the results from that analysis. “Sampleprob” (the percentage of extant species sampled in the analysis) was set to 0.005, as only extant Thryonomys (African cane rat) was sampled from the entire sample of extant hystricognaths. The root node was constrained to fall within a uniform prior on node age from 47.8 Ma (the oldest possible age of the species in the matrix) to 55 Ma (beyond which no ctenohystricans, or even demonstrable crown rodents, have been found in the fossil record; e.g., Marivaux, Vianey-Liaud & Jaeger, 2004).

Two tip-dating analyses were run: in the first analysis (referred to as TD1), each tip was calibrated with a uniform prior on age, employing minimum and maximum estimates based on the currently accepted upper and lower bounds of magenetochrons or geological stages or ages to which fossils have been assigned (i.e., in MrBayes, calibrate taxon = uniform(minimum age, maximum age); see Appendix S1, which provides justification for the minimum and maximum ages for each taxon). Fourteen of the species in the analysis are from the Fayum succession, and we followed the magnetostratigraphic correlation of the Fayum beds to the Geomagnetic Polarity Timescale that was preferred by Seiffert (2006) and Seiffert et al. (2008). In addition to topology and attendant support and parameter estimates, this first analysis importantly also output point age estimates for each tip species from within its uniform prior, taking into account (among other things) the base clockrate and the amount of change expected along the terminal branch leading to the tip. However as would be expected given such parameters, the point age estimates for species from a single locality were not all the same, as most are assumed to be, given that they are from the same stratum or tightly constrained interval (also assuming that time-averaging in an accumulation is negligible). Because tips from the same locality should ideally “line up” so that branch lengths are not artificially long or short (thereby implying artificially slow or fast rates of evolution), a second analysis (TD2) was run with the point age estimates for species from each locality (i.e., the estimates calculated in TD1) averaged and used as fixed dates [i.e., in MrBayes, calibrate taxon = fixed(mean age for locality based on TD1 estimates)]. For both analyses, the MCMC chains were run for 50 million generations, with two independent runs, each with one cold chain and three heated chains (temp = 0.01), sampling every 1,000 generations. The first 25% of the resulting 50,000 samples were discarded as the “burn-in” period, and the remaining trees were summarized using an “allcompat” (majority-rule plus compatible groups) consensus tree.

Bayesian ancestral reconstruction of first lower molar size in early hystricognaths

We collected length and width measurements on the first lower molars of early hystricognaths in our character-taxon matrix, either directly (in the case of Fayum species) or from published data (in the case of species for which we only had casts), with the goal of reconstructing the evolution of first lower molar area (natural log; see Dataset S4) within a Bayesian context using the Continuous module in BayesTraits v. 2 (Pagel, 2002; Pagel & Meade, 2013). We used the “allcompat” consensus derived from the tip-dating analysis (TD2) described above as the input tree because that analysis provided divergence dates among extinct taxa that are based on their tip ages and inferred evolutionary rates, which we considered to be preferable to arbitrarily time-scaled trees derived from parsimony or Bayesian analyses that do not take into account evolutionary rates or amount of morphological change along internodes beyond that which is minimally required by fossil ages. We first ran 10,000,000-generation MCMC analyses of the data set under random walk and directional models, with and without the phylogenetic scaling parameters delta, kappa, and lambda (each considered individually), to determine which model/scaling parameter combination had the highest log likelihood (based on the harmonic mean in the final MCMC generation). For both the random walk model and the directional model, the lambda scaling parameter was favored, but the random walk model with the lambda scaling parameter was only weakly favored over the directional model with the lambda scaling parameter (based on log Bayes factors), so we present results from both of those analyses. Model files were created by running MCMC chains for 10,050,000 generations, with the first 50,000 discarded as burn-in. These model files were then employed in longer (20,050,000 generations, first 50,000 discarded as burn-in) MCMC chains for ancestral state reconstructions, in which ancestral values were estimated for all internal nodes in the tree.

Results

Systematic paleontology

Class MAMMALIA Linnaeus, 1758	
Order RODENTIA Bowdich, 1821	
Infraorder HYSTRICOGNATHI Tullberg, 1899	
Parvorder PHIOMORPHA Lavocat, 1962	
Family Incertae sedis	

Birkamys, new genus (Figs. 3–5 and Table 1) urn:lsid:zoobank.org:act:D8D841E7-55E1-4F3A-9B41-CE37F8B6F3A7

Figure 3 CGM 66000, holotype cranium of Birkamys korai, new genus and species, from the latest Eocene Locality L-41, Jebel Qatrani Formation, Fayum Depression, northern Egypt.

Volume renderings of high-resolution micro-CT scans of CGM 66000, in (A) dorsal view, (B) ventral, (C) anterior, (D and E) lateral views. Occlusal surface of the upper teeth is shown in (F).

Figure 4 Maxillae and upper dentition of Birkamys korai, new genus and species, from Quarry L-41.

(A)–(D) Right partial maxilla of DPC 17457 with dP3–M3, in (A) dorsal, (B) ventral, (C) anterior views and (D) occlusal surface; (E)–(H) DPC 9276, left partial maxilla with M2−3 and alveoli for dP3−4 and M1, in (E) ventral, (F) anterior, (G) dorsal views and (H) occlusal surface; DPC 15625, left partial maxilla with M1 and alveoli for dP3−4, in (I) dorsal, (J) anterior, (K) ventral views and (L) occlusal surface.

Figure 5 Mandible and lower dentition of Birkamys korai, new genus and species, from Quarry L-41.

(A–F) DPC 22737, left mandible with dP4–M3, in (A) medial, (B) lateral, (C) ventral and (D) dorsal views; (E) scanning electron stereopair illustrating the occlusal surfaces of dP4–M3; (F) cross-section through tooth roots, illustrating the deep roots of the dP4 in cross-section.

Table 1 Length and width of teeth (in millimeters) in the hypodigm of Birkamys korai, cf. Birkamys, and Mubhammys vadumensis

Birkamys korai	
Specimen no.	Side	L	W	L	W	L	W	L	W	L	W	
Upper teeth		dP3	dP4	M1	M2	M3	
CGM 66000 (Holotype)	Left	0.36	0.33	1.03	1.03	1.07	1.13	1.00	1.23	0.87	0.80	
Right	0.27	0.30	1.07	1.07	1.07	1.13	1.00	1.23	0.83	1.07	
DPC 17457	Right	0.30	0.30	0.97	1.03	1.06	1.10	0.93	1.13	0.83	9.7	
DPC 15625	Left	–	–	–	–	1.06	1.03	–	–	–	–	
DPC 9276	Left	–	–	–	–	–	–	1.08	1.25	0.95	1.10	
Lower teeth				dP4	M1	M2	M3	
DPC 22737	Right			1.10	0.76	1.03	0.93	1.10	1.06	1.03	0.96	
Cf. Birkamys korai	
Specimen no.	Side					M1					
DPC 17995	Right					1.10	0.87					
Mubhammys vadumensis	
Specimen no.	Side											
Upper teeth		dP3	dP4	M1	M2	M3	
CGM 66001 (Holotype)	Left	0.68	0.76	1.8	1.9	1.90	–	1.95	2.10	–	1.80	
DPC 14324	Left	0.58	0.68	1.70	1.60	1.85	1.80	–	–	–	–	
Lower teeth				dP4	M1	M2	M3	
DPC 13220	Left			1.83	1.43	2.00	1.75	1.95	1.80	1.83	1.70	
DPC 14141	Left			1.66	1.29	1.90	1.67	–	–	–	–	

Type and only known species

Birkamys korai, new species urn:lsid:zoobank.org:act:CF09DD3A-E265-43FA-85B6-E9868F9B8364

Etymology

Combination of birka, Arabic word for lake or swamp, in reference to the L-41 deposits and mys, Greek for mouse.

Diagnosis

As for the type and only known species.

Birkamys korai, new species urn:lsid:zoobank.org:act:CF09DD3A-E265-43FA-85B6-E9868F9B8364 (Figs. 3–5 and Table 1)

Etymology

In honor of Professor Mahmoud Kora of Mansoura University, for his important contributions to the study of stratigraphy and paleontology in Egypt.

Holotype

CGM 66000, rostrum with right and left upper incisors and dP3-M3 (Fig. 3; measurements in Table 1).

Referred specimens

DPC 9276, left maxilla with M2 and M3 (Figs. 4E–4H); DPC 15625, left maxilla with M1 (Figs. 4J–4L); DPC 17457, right maxilla with dP3-M3; DPC 22737, right mandible with dP4-M3 (Fig. 5).

Type locality

Locality 41 (L-41), 46 m above the base of the lower sequence of the Jebel Qatrani Formation. The fine green claystone at L-41 contains 12% postdepositional sodium chloride and is unique among Fayum fossil quarries, most of which occur in sands and gravels. The sediments at L-41 might have been laid down in a freshwater lake that was periodically flooded, resulting in large accumulations of vertebrate carcasses (Simons, Cornero & Bown, 1998). Over the last three decades, work at L-41 has produced a wide variety of fish, amphibian, reptile, bird and mammal taxa. There is no clear sorting of fossil mammals on the basis of size, and the locality contains not only large-bodied hyracoids (Rasmussen & Gutiérrez, 2010) and anthracotheriid artiodactyls, but also very small primates (Simons, 1997; Simons et al., 2001; Seiffert et al., 2005), macroscelideans (Simons, Holroyd & Bown, 1991), tenrecoids (Seiffert & Simons, 2000; Seiffert et al., 2007), bats (Gunnell, Simons & Seiffert, 2008), and rodents (Holroyd, 1994; Lewis & Simons, 2006; Sallam, Seiffert & Simons, 2011; Sallam, Seiffert & Simons, 2012). Hundreds of rodent specimens are known from L-41, but the only clade represented is Hystricognathi, whereas both Hystricognathi and Anomaluroidea occur at the ∼37 Ma Locality BQ-2 (Sallam et al., 2009; Sallam, Seiffert & Simons, 2010a; Sallam et al., 2010b).

Age and Formation

Latest part of late Eocene (latest Priabonian, ∼34 Ma), lower sequence of Jebel Qatrani Formation, northern Egypt.

Diagnosis

Birkamys korai differs from early Oligocene Phiomys andrewsi in having smaller molars; in showing no evidence for eruption of permanent premolars; in having relatively short metalophulid IIs on dP4-M3; in having a relatively small M3 when compared to M2; and in lacking a mesostyle and mesolophule on the upper molars. Differs from contemporaneous and sympatric Acritophiomys bowni in having smaller teeth; in showing no evidence for replacement of deciduous premolars; in having a relatively weak anterior cingulid, no metalophulid I or II, and no mesolophid or mesostylid on dP4; in having no hint of an M1−2 anteroconid; in consistently lacking M1−2 mesostylids and mesolophids, and having relatively short metalophulid IIs; in having relatively large M1−2 protoconids; in having a relatively small M3 when compared to M2; in lacking a dP4 mesolophule; in consistently lacking M1−2 mesostyles, mesolophules, and pericingula; in having relatively large M1−2 metaconules; and in lacking enamel wrinkling and crenulation. Differs from Oligocene Turkanamys hexalophus from Kenya in having smaller molars; in showing no evidence for replacement of deciduous premolars; in having relatively well-developed anterior cingulids on M1−2; in having a small, rather than large, metastylid on M1−2; in lacking mesostylids and well-developed metalophulid IIs on M1−2; in lacking a connection of the entoconid and hypoconid on the M1−2; in having a relatively small M3 when compared to M2; in lacking M1−2 mesostyles and mesolophules; and in lacking enamel wrinkling and crenulation. Differs from “Phiomys” hammudai from the late Eocene of Libya in having smaller molars; in showing no evidence for replacement of deciduous premolars; in lacking a dP4 mesolophid, mesostylid, metalophulid I, and metalophulid II; in having a more distinct anterocingulid but lacking a mesostylid, mesolophid, and a well-developed metalophulid II on M1−2; in having a relatively small M3 when compared to M2; in lacking M1−2 mesostyles and mesolophules, and having relatively large metaconules; and in lacking enamel wrinkling and crenulation. Differs from early Oligocene Neophiomys paraphiomyoides from Egypt and Libya in lacking a distinct metalophulid II and having a complete ectolophid on dP4; in having a relatively small M3 when compared to M2; in lacking M1−2 mesostyles, mesolophules, and posterior arms of paracones; and in having relatively large M1−2 metaconules. Differs from early Oligocene “Phiomys” lavocati from younger quarries in the Fayum succession in showing no evidence for replacement of deciduous premolars; in having a dP4 protoconid that is more mesially placed relative to the metaconid; in having M1−2 protoconids that are relatively large when compared with metaconids; in having a dP4 metaloph that is connected to the metaconule, rather than distally oriented; in having a dP4 mure; in having M1−2 mures, and metaconules that are submerged into the mures; in having anterior arms of the M1−2 hypocones that are relatively well-developed; in having an M1 metaloph that is connected to both the metaconule and posteroloph; and in lacking posterior arms of M1−2 paracones. Differs from late Eocene Talahphiomys lavocati from Libya in having a dP4 protoconid that is more mesially placed relative to the metaconid; in lacking a dP4 mesostylid; in having a more distinct M1−2 anterocingulid; in having a dP4 metaloph that is connected to the metaconule, rather than distally oriented; in having a dP4 mure; in lacking an M1−2 mesostyle; in having M1−2 mures and metaconules that are submerged into the mures; and in having an M2 metaloph that is connected both to the metaconule and the posteroloph. Differs from late Eocene Talahphiomys libycus from Libya in having a dP4 metaloph that is connected to the metaconule, rather than distally oriented; in having a dP4 mure; in lacking an M1−2 mesostyle and mesolophule; in having an M2 metaloph that is connected both to the metaconule and the posteroloph; in having M1−2 mures; and in lacking posterior arms of the M1−2 paracones and anterior arms of the M1−2 metacones.

Description

Four crushed cranial elements of Birkamys korai (Figs. 3 and 4) together document much of the morphology of the rostrum, mid-cranium, and the complete upper dentition. The holotype rostrum CGM 66000 (Fig. 3) was subjected to severe post-mortem distortion that has led the specimen to be dorsoventrally flattened with numerous surface cracks and displacements; rather than attempt physical preparation of this tiny and very fragile specimen, we figure as much as is possible through volume rendering of the encasing block using high-resolution micro-CT scans (Fig. 3). CGM 66000 contains the premaxillae with two upper incisors, both maxillae with the entire dentition (dP3-M3), and most of the frontal. The cranial parts in the hypodigm represent adult individuals, two of which preserve third and fourth deciduous premolars that are worn.

The premaxillae are preserved in the holotype, house the two upper incisors, and form most of the mediolaterally narrow and anteroposteriorly elongate rostrum and upper diastema (Fig. 3). The most striking feature of the rostrum is the capacious vacuity, referred to by some as an “anterior palatine fenestra” (e.g., Wood, 1968), formed by the anteroposteriorly elongate and confluent incisive foramina, the anterior halves of which deeply excavate, and are formed by, the premaxillae. The posterior border of the incisive foramen extends posteriorly between dP3 and dP4. The suture between the premaxilla and the maxilla is well-preserved. Birkamys was hystricomorphous; that is, the infraorbital foramen was very large and presumably allowed for the passage of a greatly expanded medial masseter that inserted on the side of the rostrum, anterior to the zygomatic arch. The size and shape of the ventral margin of the infraorbital foramen is most clearly preserved on DPC 9276 (Fig. 4F). On the ventral surface of the maxilla, a small masseteric tubercle is situated immediately ventral to the infraorbital foramen and anterolateral to the alveolus of dP3, providing a point of origin (along with the zygomatic arch, which extends laterally at the level of the alveolus of dP3 and masseteric tubercle) for the superficial masseter. On the dorsal view of DPC 9276, the infraorbital fissure is relatively broad and deepens anteriorly, separating the orbital process from the alveolar portion (Fig. 4G). The alveolar foramen is oval in shape and lies within the medial wall of the infraorbital fissure, dorsal to the dP3 alveolus. The anterior portion of the maxilla protrudes roughly at the same level as the alveolar plane and preserves part of the articulation with the premaxilla. The palate is preserved in the holotype and is somewhat flat, slightly lower than the alveolar plane, and broad throughout its length. It houses the two major palatine foramina, which are relatively round and large, and which lie at the level of the first upper molar.

The upper deciduous third premolar (dP3) is generally a small peg-shaped tooth with a rounded base, and abuts the mesial surface of dP4. The dP3, on both the left and right sides, is well-preserved in the holotype and DPC 17457 (Figs. 3F and 4D). It has one large cusp that occupies the distal portion of the crown and forms the apex of the tooth. There is a small depression on the distolabial side of the former cusp.

The upper deciduous fourth premolar (dP4) is roughly quadrate in shape and bears four major cusps (paracone, metacone, protocone and hypocone) as well as a small metaconule (Figs. 3F and 4D). The paracone is of approximately the same size and height as the metacone, and the hypocone is situated at the same level as the protocone. The occlusal configuration of the crown is tetralophodont, with no mesolophule. The anteroloph that forms the anterior border of the tooth is low and runs labially from the protocone, parallel to the protoloph, and fuses with a weakly-developed parastyle just mesial to the base of the paracone. A well-developed and transversely oriented protoloph joins the paracone and protocone. There is a small anterostyle at the junction of the protocone and anteroloph. The metaloph takes a sinuous course, running lingually and mesially from the metacone, connecting to the metaconule. There is a weak connection with the posteroloph at its lingual portion, which, together with the hypocone and its anterior arm, form a small fovea at the distolingual corner of the crown. The posteroloph runs labially from the hypocone, courses around the posterior margin of the tooth and connects to the distal base of the metacone. A weak and poorly defined metaconule is centrally placed and connected to the hypocone via the latter cusp’s anterior arm, which is robust and oriented mesiolabially. The mure is complete, connecting the metaconule with the mid-point, or lingual part of, the protoloph. The labial wall, which is formed primarily by the posterior arm of the paracone, bears a deep notch that extends to the level of the central basin, and there is a very weakly developed swelling that could be a remnant of a mesostyle. The tooth lacks an endoloph, and the lingual sinus (=hypoflexus) is wide, deep, and is not continuous with the central basin (=mesoflexus) due to the presence of the mure.

The upper first molar (M1) is the largest tooth of the upper dentition (Figs. 3F, 4D and 4L). Its occlusal pattern is nearly identical to that of dP4, but differs in having relatively tall lophs and cusps that are completely integrated into the four primary crests (anteroloph, protoloph, metaloph, and posteroloph), and in having a relatively tall paracone when compared to the metacone. The M2 occlusal surface is similar to that of M1, differing only in being relatively shorter and broader. The hypocone is more labially situated with respect to the protocone.

The upper third molar (M3) is smaller than M1−2 and has a relatively short lingual margin, leading to a somewhat oval outline (Figs. 3F, 4D and 4H). The tooth bears a reduced metacone and hypocone, which are relatively lingually and labially positioned, respectively, when compared with the same cusps on M1,2. The metaloph is weakly developed and connects the metacone and the anterior arm of the hypocone. The central basin is closed by a weakly developed labial wall, forming a mesofossette, and is closed lingually by a weak to well-developed mure that reaches the lingual aspect of the protoloph. On DPC 17457, an outgrowth from the posterior part of the protocone extends distolabially toward the anterior arm of the hypocone, forming a high and continuous neo-endoloph. Together with the mure, the protoloph and neo-endoloph delimit a small fovea. On DPC 9276, there is a short and low anterior arm of the hypocone that is connected to the base of the metaconule. The latter is relatively well-developed and more distal in position when compared with the same cusp on M2. There is a small crest running longitudinally from the base of the posteroloph to connect with the metaconule distally. This crest contributes to a small fovea that is also bordered by the metaconule, the hypocone and its anterior arm, and the lingual part of the posteroloph. The posteroloph is relatively well-developed with respect to that of M2, and courses around the posterior margin of the tooth, running labially from the hypocone to form a strong connection with the metacone.

The mandible (Figs. 5A–5D) is slender, with a partially preserved ascending ramus and a well-preserved corpus. The angular process is placed lateral to the tooth row and the incisor, leaving a well-developed groove between the angular process and the incisor alveolus; the mandible is thus fully hystricognathous. On the lateral surface of the mandible, the ventral masseteric ridge extends laterally, arising near the midpoint of the horizontal ramus and continuing posteriorly toward the angular process; this ridge, which is an insertion site for the deep masseter muscle, extends anteriorly as a part of the masseteric fossa, and terminates beneath the posterior portion of the dP4 at roughly the same level as the mental foramen. The dorsal masseteric ridge is less developed anteriorly and extends posteriorly with the coronoid process. The tip of the coronoid process is broken but it is apparently higher than the condylar process; it arises lateral to the third molar, leaving a deep fossa. The mental foramen is somewhat oval and can be seen in dorsal view, lying anterior to the anterior root of the dP4 and ventral to the distal portion of the diastema. The diastema is deep and makes up about half the length of the horizontal ramus. The posterior part of the ascending ramus is extremely fragile, and the tip of the angular process is not preserved. On the medial surface of the mandible, the mandibular foramen is situated in the area between the coronoid and condylar processes, on the dorsal margin of a strut that extends posteriorly from the rear part of the incisor alveolus. The condylar process is slightly higher than the tooth row and the tip of the incisor (which are roughly at the same level). The lower incisor is well-preserved; it is oval in cross–section, with somewhat flat medial and lateral margins. The anterior surface of the incisor is covered by smooth enamel that extends only to the labial surface, covering about one-third of the labial side of the incisor.

The lower deciduous premolar (dP4) is slightly less worn than the lower molars (Fig. 5). A micro-CT scan revealed no hint of a developing p4 (Fig. 5F), suggesting that Birkamys likely did not replace dP4—a condition that occurs in later-occurring phiomorphs aside from Phiomys. The dP4 is longer than it is wide, and has a somewhat rectangular outline with a wide talonid and a narrow trigonid. The tooth bears five major cusps (metaconid, entoconid, protoconid, hypoconid and hypoconulid). The lingual cusps are slightly smaller than the labial cusps, and the hypoconulid is the smallest cusp. The protoconid extends distal to the metaconid, and has a short crest running mesially from its mesiolingual portion that meets the anteroconid. A low and weakly developed anterocingulid extends around the mesial margin of the tooth. The middle portion of the crown is open mesially, due to the absence of the metalophulid I and II, and open lingually due to the absence of the anterior arm of the entoconid and the short posterior arm of the metaconid. The ectolophid is low and attaches to the anterior arm of the hypoconid near that crest’s junction with the hypolophid. The entoconid is placed mesial to the hypoconid. The hypoconid is connected to a distinct hypoconulid by a well-developed posterolophid that runs across the distal border of the crown and does not reach the distal face of the entoconid, leaving the posterior basin (=metaflexid) open lingually. The labial sinusid (=hypoflexid) is wide and shallow with no ectostylid.

The first lower molar (M1) is somewhat rectangular in outline and relatively broad, and bears long and relatively well-developed lophs when compared with those of the dP4 (Fig. 5). The mesial wall of the tooth is formed by a trenchant and concave metalophulid I that connects the protoconid and metaconid. A low anterocingulid is present mesial to the protoconid. A short posterior arm of the protoconid protrudes lingually; thus the tooth bears only two major basins. The ectolophid is more lingually positioned than that on dP4. The posterior arm of the metaconid tapers and ends near the midline of the tooth, leaving the anterior basin (=mesoflexid) open lingually via a narrow valley. The hypoflexid is broad and deep.

The second lower molar (M2) is the largest tooth of the lower dentition. The occlusal configuration is identical to that of the first molar, but the tooth is relatively broad. The anterior margin is straighter and the anterocingulid is less developed than on the M1 and dP4. The occlusal pattern on the third molar is also very similar to that of M1−2, but differs in having a relatively weak anterocingulid, a trigonid that is slightly broader than the talonid, a relatively well-developed posterior arm of the protoconid, and a hypoconulid that is submerged into the posterolophid, forming the rear lobe of the crown.

Cf. Birkamys (Fig. 6)

Figure 6 Mandible of cf. Birkamys from Quarry L-41.

(A–E) DPC 17995, almost complete right mandible with I and M1, in (A) lateral, (B) medial, (C) ventral and (D) dorsal views; (E) occlusal surface.

An almost complete hemi-mandible (DPC 17995) has been recovered from L-41 that preserves the lower incisor and M1 (Fig. 6). There are few differences from the mandible of Birkamys korai (DPC 22737) in the morphology of the corpus, ascending ramus, and in the position and development of the masseteric crests, but the anterior border of the ascending ramus begins to curve dorsally just posterior to the distal aspect of M1, whereas on DPC 22737 the dorsal curvature begins at about the mid-point of M3; therefore it seems likely that the ascending ramus would have obscured M3 in lateral view on DPC 17995. The M1 preserved in DPC 17995 also differs from that in DPC 22737, notably in being longer relative to width (1.26 times longer than wide, as opposed to 1.10 times longer than wide); having a distinct, but short, posterior arm of the protoconid; in having a lower crown height and relatively thin crests; in having a more broadly open hypoflexid; and in having a metalophulid I that is mesially convex. We refrain from naming a new taxon based on this material because only one diagnostic tooth is known, but the differences between DPC 17995 and DPC 22737 suggest that this might be an additional tiny new species that could even be distinct at the generic level.

Mubhammys, new genus (Figs. 7, 8 and Table 1) urn:lsid:zoobank.org:act:BAE3043A-8B71-441F-B0C3-9A0B7E6CC630

Figure 7 Maxillae and upper dentition of Mubhammys vadumensis, new genus and species, from the latest Eocene Quarry L-41, Jebel Qatrani Formation, Fayum Depression, northern Egypt.

(A–B) CGM 66001, holotype left partial maxilla (upper left incisor, dP3−4 and M1−3) in (A) medial and (B) occlusal views; (C–H) DPC 14324, left partial maxilla and upper dentition (I, dP3−4 and M1) in (C) ventral, (D) anterior and (E) dorsal views; (F and G) medial and ventral surface of upper left incisor; (H) occlusal surface.

Figure 8 Lower dentition of Mubhammys vadumensis, new genus and species.

(A–D) Left mandibular fragment and lower dentition DPC 14141 (dP4–M1), (A) occlusal surface, (B) lateral, (C) dorsal and (D) medial views; (E–H) left mandibular fragment and lower dentition DPC 13220 (dP4–M3), (E) occlusal surface, (F) lateral, and (H) medial views and (G) micro-CT scans, illustrating the deep roots of the dP4 in cross-section.

Type and only known species

Mubhammys vadumensis, new species urn:lsid:zoobank.org:act:1E6685EA-BAF3-4035-B62D-0D0C486A3D46

Etymology Combination of mubham, Arabic word for enigmatic or mysterious, and mys, Greek for mouse.

Diagnosis

As for the type and only known species.

Mubhammys vadumensis, new species urn:lsid:zoobank.org:act:1E6685EA-BAF3-4035-B62D-0D0C486A3D46 (Figs. 7, 8 and Table 1)

Etymology

From Greek vadum for shallow, in reference to the depositional environment of L-41.

Holotype

CGM 66001, a left maxilla with dP3−4 and M1−3 (Figs. 7A and 7B, measurements in Table 1).

Referred specimens

DPC 14324, left maxilla with dP3–M1 (Figs. 7C–7H), DPC 13220, left mandibular fragment with dP4–M3 (Figs. 8E–8H), DPC 14141, left mandibular fragment with dP4–M1 (Figs. 8A–8D).

Type locality

Locality 41 (L-41).

Age and Formation

Latest part of late Eocene (latest Priabonian, ∼34 Ma), lower sequence of Jebel Qatrani Formation, northern Egypt.

Diagnosis

Relatively large basal phiomorph that shows no evidence for replacement of deciduous premolars. Differs from Birkamys korai primarily in its larger size, but also in lacking a dP4 anteroconid; in having metalophs on dP4–M3 that are submerged into posterolophs; and in having small M1−2 mesostyles. Differs from contemporaneous and sympatric Acritophiomys bowni in showing no evidence for replacement of deciduous premolars; in lacking a distinct anteroconid, mesostylid, mesolophid, metalophulid I, and metalophulid II on dP4; in lacking a mesolophid, metalophulid II, and incipient anteroconid on M1−2; in having relatively large lower molar protoconids when compared to metaconids; in having M3∕3 relatively small when compared with M2∕2; in lacking mesolophules on dP4-M2; in having relatively large M1−2 metaconules; in having M1−3 metalophs that are submerged into the posterolophs; and in lacking enamel wrinkling and crenulation on molars. Differs from younger Metaphiomys beadnelli, also from the Fayum succession, in lacking a dP4 anteroconid; in lacking a metalophulid II on dP4; in having a larger dP4 hypoconulid; in having relatively large lower molar protoconids when compared to metaconids; in having M3∕3 relatively small when compared with M2∕2; in having a relatively low dP4 anterocingulum; in having dP4–M3 metalophs that are submerged into posterolophs; in lacking mesolophules on dP4–M2; in having relatively large M1−2 metaconules; and in having no posterior arm of the paracone on M1−2. Differs from Oligocene Turkanamys hexalophus from Kenya in showing no evidence for replacement of deciduous premolars; in having a low anterocingulid on M1−2; in having small metastylids, no mesostylids, and very weak posterior arms of the protoconid on M1−2; in having relatively large lower molar protoconids when compared to metaconids; in having no connection of the entoconid and hypoconid via the posterolophid on M1−2; in having M3∕3 relatively small when compared with M2∕2; in having small parastyles, weak anterostyles, and no mesolophule on M1−2; in having metalophs submerged into posterolophs on M1−3; and in lacking enamel wrinkling and crenulation on the molars.

Description

On the medial surface of the broken premaxilla of CGM 66001 (Figs. 7F–7G) a complete right upper incisor is exposed; it is short and highly arched when compared with the lower incisor. It is oval in occlusal outline, with a flat medial surface and curved dorsal and lateral surfaces. In lateral view, the occlusal surface is longer when compared with that of the lower incisor. The pulp cavity is short and slit-shaped, and placed at the middle of the occlusal surface. A smooth enamel layer covers the mesial surface of the upper incisor and extends labially to cover only one-third of the labial side.

The maxillary morphology of Mubhammys appears to be very similar to that of Birkamys, with a similar placement of the zygomatic process, and an anteroposteriorly extensive concave margin of the lateral border of the incisive foramen. As in Birkamys, the posterior margin of the incisive foramen would have extended posteriorly to approximately the point of DP4, and the anterior margin presumably extended far into the premaxilla. Mubhammys thus shares the remarkably enlarged incisive foramen morphology that is seen in Birkamys.

The upper deciduous third premolars (dP3) are preserved in the holotype and DPC 14324. It is a small peg-shaped tooth with a primary cusp and a rounded base, and abuts the mesial surface of dP4. The dP3 is less worn than the upper molars (Figs. 7A–7D and 7H).

The upper deciduous fourth premolar (dP4) has a somewhat trapezoidal outline, with a relatively short lingual margin and a broad labial margin (Figs. 7A–7C and 7H). The occlusal surface has four enlarged cusps (paracone, protocone, metacone, and hypocone), all of which are about equal in size and height. The mesostyle is a distinct and isolated cusp and is situated midway between the paracone and metacone along the buccal margin of the tooth. A small anterostyle is situated mesiolabial to the protocone, midway along the mesial margin of the tooth. The anteroloph is low and short, runs labially from the anterostyle, and courses across the mesial margin of the tooth. It continues distally to end labially to the metacone, forming a cingulum around the mesiolabial corner of the crown. The protoloph is a very short transverse crest, runs labially from the protocone, and flares labially to meet the paracone, which is large and lacks a posterior arm. Due to the absence of the posterior arm of the paracone and the anterior arm of the metacone, the mesoflexus is open labially via a wide notch. The hypocone is well-developed and placed distal to the protocone. The anterior arm of the hypocone is robust and runs mesiolabially to end at the middle of the tooth. There is a very low and weakly developed mure, connecting the base of the anterior arm of the hypocone with the protoloph. The mesolophule is absent. The metaloph is robust but very short and its lingual end is submerged into the posteroloph, the latter of which runs labially from the hypocone and ends at the distal base of the metacone. The posterior basin (=posteroflexus) is shallow and very narrow when compared with the anterior basin (=paraflexus). Distal to the anterostyle and mesial to the protocone, there is a short and low anterocingulum. The labial sinus is deep and narrow. There is no ectostyle.

The upper first molar (Figs. 7A–7C and 7H) has a similar occlusal morphology to that of dP4, but it is larger, and the lophs and cusps are relatively well-developed. The outline of the tooth is roughly square. The anteroloph has a relatively straight course and is strongly connected lingually to a well-developed protocone, unlike that of dP4. The metaloph is relatively long with respect to that of dP4 and turns backward where it is confluent with the posteroloph. The mesostyle is relatively well-developed and extends toward the mesoflexus via a short strut. In the holotype, the metacone is broken.

The upper second molar is only preserved in the holotype (Figs. 7A and 7B) and it is strongly worn, but in general appears to be similar in morphology to M1. It differs in being larger and having a reduced and more labially and lingually situated hypocone and metacone (broken), respectively. The metaloph is relatively long with respect to that of M1. The hypoflexus is relatively wider when compared with those of dP4 and M1. The cingulum around the mesiolabial corner of the tooth is absent.

The upper third molar (Figs. 7A and 7B) is heart-shaped and is smaller than all of the other upper teeth aside from dP3. The anterior half of the tooth is similar to the corresponding part on M1−2, but the posterior part is very worn and the metacone is broken. The tooth has a large and more crestiform protocone, occuping most of the lingual portion of the crown. The hypocone is very small and is relatively labial in position with respect to the protocone. The course of the metaloph is uncertain due to wear.

The mandible is slender and hystricognathous, with its angular process placed lateral to the plane of the incisor and the tooth row, leaving a well-developed groove between the angular process and this plane (Figs. 8B–8D and 8F–8H). DPC 13220 (Figs. 8E–8H) represents a fragment of a left mandibular corpus with dP4-M3 and the middle part of the incisor. On the labial surface of the specimen, the masseteric fossa is pronounced, ending beneath the first molar. The ventral masseteric ridge extends farther mesially below the dP4 and is located lateral to the incisor and teeth row. The dorsal masseteric ridge is less developed when compared to the ventral ridge and extends distally along the base of the coronoid process. The groove between the tooth row and the base of the coronoid process is preserved in DPC 13220 (Figs. 8F and 8H). The lower incisor is oval with somewhat flat medial, and slightly convex lateral, margins. A smooth and thick enamel layer covers the ventral surface of the incisor and extends on both the mesial and distal sides. The pulp cavity is elongate in outline and sits in the middle of the dentine layer. The mental foramen is oval-shaped and relatively small and is located between the distal portion of the diastema, slightly above the level of the mesial end of the ventral masseteric ridge. The diastema is well-preserved in specimen DPC 14141; it is slightly deeper than the alveolar row and is as long as the length of the two first teeth. The tip of the angular process, coronoid process, and mandibular process are not preserved.

The two mandibular fragments of Mubhammys vadumensis (DPC 13220 & DPC 14141) preserve lower cheek teeth (Fig. 8). The dP4 is longer than it is wide, and has a somewhat oval outline with a wide talonid and a narrow trigonid. The tooth bears a very weakly developed anterocingulid mesial to the protoconid. The metalophulid I and the posterior arm of the protoconid are absent, leaving the anterior basin (=anteroflexid) open mesially via a deep and narrow notch. The metaconid is larger than the protoconid and is more mesially positioned. The hypoconid is placed slightly posterior to the entoconid. A short anterior arm of the hypoconid attaches to the hypolophid, which flares lingually along the apex of a large entoconid. The ectolophid is low relative to cusp height, and joins the protoconid at the junction of the anterior arm of the hypoconid and the hypolophid. The posterior arm of the metaconid slopes distally as a part of the lingual wall but terminates before the midline of the crown. The anterior arm of the entoconid is absent, leaving a wide notch along the lingual side of mesoflexid. There is no mesoconid or mesolophid. The posterolophid is well-developed, running distolingually from the hypoconid, coursing around the posterior margin of the tooth to end distal to the entoconid, leaving a wide opening on the lingual margin of the metaflexid. A well-developed hypoconulid occupies the middle portion of the posterolophid and forms the very distal tip of the tooth. A low, poorly developed postcingulid runs labially from the distal end of the hypoconulid. The hypoflexid is wide and deep. In DPC 14141, dP4 has a well-developed cusp that abuts the metaconid distally, while the mesostylid is represented by a small cuspid.

The first lower molar is roughly rectangular in outline and bears relatively well-developed lophs and cusps; the mesial part of the tooth is slightly narrower than the distal part. An incipent and low anterocingulid runs parallel to the metalophulid I; it is relatively well-developed when compared with that on dP4. The metalophulid I is well-developed, forming the mesial border of the tooth, and runs from the anterior side of the protoconid toward the mesial side of the metaconid; it bears a shallow notch near its labial one-third. The posterior arm of the protoconid is represented by a very small knob on the mesial part of the ectolophid. The latter is well-developed and higher than that of the dP4, and connects the protoconid to the junction of the hypolophid and the anterior arm of the hypoconid. The posterior arm of the metaconid is relatively short when compared with that of the dP4, and only forms about one-third of the lingual wall between the metaconid and the entoconid. The anterior arm of the entoconid is absent, leaving the mesoflexid open via a lingual notch that is narrower than that on dP4. The hypoflexid is relatively narrow, and the postcingulid that runs labially from the distal end of the hypoconulid is relatively well-developed with respect to that on dP4.

The occlusal surface of M2 (Figs. 8E–8H) is very similar to that of M1. The M2 differs in being relatively wider and shorter, and in having a slightly broader trigonid and a longer talonid. The notch of the metalophulid I is relatively deep when compared with that on M1, but still higher than the anterior basin. The posterior arm of the protoconid is very short and is oriented toward the lingual wall of the tooth, but it is relatively long when compared with that on M1. The hypoconulid and its postcingulid are not as well-developed at they are on dP4-M1.

The mesial portion of M3 (Figs. 8E–8H) is somewhat similar to the corresponding part of M1 and M2. The M3 differs from the rest of the lower dentition in being relatively small, and in having a triangular outline. The trigonid is relatively large when compared with the talonid. The posterior arm of the protoconid is relatively long and is oriented toward the entoconid, which is reduced in size. The hypolophid is short and the hypoconid flares with the posterolophid distally, forming the posterior margin of the tooth. The hypoconulid is absent. The anterior arm of the entoconid is present, leading to a narrow and deep notch on the lingual wall; the mesoflexid is closed lingually, forming a mesofossettid. The postcingulid is absent, and the anterior arm of the hypoconid is very short.

Comparisons

The contemporaneous and sympatric L-41 species of Birkamys and Mubhammys are quite different in tooth and mandible size (Fig. 9), but are very similar in occlusal morphology. B. korai is the smallest known hystricognathous rodent from the Paleogene of Africa, and the first lower molar area of M. vadumensis is 3.5 times that of B. korai. The lower dentition of Birkamys is similar to that of Mubhammys in lacking metalophulid I and II on dP4, and in having a short posterior arm of the protoconid on M1 and a more robust posterior arm of the protoconid on the last two molars that never reaches the lingual wall. Moreover the upper dentition of Birkamys is similar to that of Mubhammys in lacking the connection between the metacone and the anterior arm of the hypocone; instead the metaloph is directed distally and submerged into the posteroloph. The lower molars of Birkamys differ from those of Mubhammys in lacking the postcingulid, having a relatively robust metalophulid I on the lower molars, and in having a relatively short M1. Birkamys also lacks the M2 mesostyle that is present in Mubhammys. Furthermore, when compared with that of Mubhammys, the dP4 of Birkamys is relatively long, has a crest that runs mesially from the protoconid, has a well-developed anterocingulid, and a large protoconid when compared with the metaconid; Mubhammys has a relatively large metaconid. The M3 of Mubhammys is heavily worn, which makes it difficult to compare it with that of Birkamys.

Figure 9 Plot of length versus width of M1 comparing Mubhammys vadumensis and Birkamys korai with other hystricognaths from the Fayum Depression.

Measurements for “Paraphiomys,” Metaphiomys, Phiomys, and “Phiomys” lavocati are from Wood (1968) and Holroyd (1994); Gaudeamus spp. from Wood (1968) and Sallam, Seiffert & Simons (2011); Acritophiomys from Sallam, Seiffert & Simons (2012); Waslamys and Protophiomys are from Sallam et al. (2009).

Birkamys and Mubhammys share a number of dental features with early Oligocene members of the genera Phiomys and Neophiomys (Coster et al., 2012b; Holroyd, 1994; Wood, 1968) that were presumably present in the last common ancestor that these taxa shared with all later phiomorphs. The type species of Phiomys (Phiomys andrewsi) differs from Birkamys in being larger, having a well-developed metalophulid II, in replacing the dP4, and in having a relatively narrow anterior portion of dP4. When compared with Mubhammys, the lower teeth of P. andrewsi are smaller, with a relatively well-developed anterocingulid. The lower molars of Birkamys differ from those of Neophiomys from the early Oligocene Fayum Quarry G (Coster et al., 2012b; Wood, 1968) in their small size, in having a well-developed metalophulid I, a relatively weak anterocingulid, and a relatively long dP4 that has the anteroconid connected to the protoconid. Furthermore, the M2 of Birkamys differs from that of Neophiomys in lacking a mure, a mesolophule, and a metaloph that is connected with both the posteroloph and mesolophule. Neophiomys is similar to Mubhammys in having an M1 that is the longest tooth of the lower dentition, in the development of the posterior arm of the protoconid, and in having an anterocingulid, but differs in its small size, interrupted metalophulid I, and well-developed anterocingulids that are present on M3. Furthermore, the dP4 of Neophiomys differs from that of Mubhammys in having an anteroconid and a long posterior arm of the protoconid that forms a complete metalophulid II. The upper dentition of Neophiomys differs from that of Mubhammys in having a double (mesial and distal) connection of the metaloph on M1, as well as a well-developed mure and mesostyle. When compared to Birkamys, the lower teeth of “Phiomys aff. paraphiomyoides” from the early Oligocene Fayum Quarry I are larger, and there is a small cusp on the anterocingulid of M1−2. The dP4 of “Phiomys aff. paraphiomyoides” also shows some characters that differ from those of Birkamys, such as being relatively short and in having a complete metalophulid II and a distinct anteroconid. The lower cheek teeth of Mubhammys show a great similarity to those of “Phiomys aff. paraphiomyoides,” differing only in having a dP4 that lacks metalophulid II, and in having a relatively long M1 with deep lingual notches, relatively well-developed postcingulids, and in lacking a small cusp on the weak anterocingulid.

“Phiomys” lavocati (Wood, 1968) is roughly the same size as Birkamys korai, but has a more robust mandible that bears a more ventral position of the mental foramen, as well as a relatively short diastema. The lower teeth of “Phiomys” lavocati differ from those of Birkamys in being broader and shorter, having relatively weak metalophulid I and II, in lacking anterocingulids, and in having mesoflexids and metaflexids that are closed lingually. The lower deciduous premolar of “Phiomys” lavocati is also relatively short, and has a relatively narrow trigonid and wide talonid, a relatively well-developed ectolophid, and an anteroconid. The dP4 protoconid of “Phiomys” lavocati is more distally positioned with respect to the metaconid, and the anterocingulid is more labially placed. The upper second molar of “Phiomys” lavocati differs from that of Birkamys in having a relatively reduced and more lingually positioned metacone; a metaloph that is poorly developed and oriented toward the metaconule; a hypocone that is equal in size to the protocone and less labially postioned; an anteroloph that is relatively well-developed; a high labial wall that closes the mesoflexus labially; a mesostyle that is integrated into the labial wall, but still recognizable; an enterostyle that abuts the hypocone, and a mure extending from the metaconule to contact the posterior portion of the protoloph.

The lower molars of “Phiomys” cf. lavocati from the early Oligocene of Oman (Thomas et al., 1989) are similar to those of Birkamys in having an anterocingulid and short posterior arm of the protoconid, and a lingually closed mesoflexid on M3. But the only upper molar from that locality shows significant differences when compared with that of Birkamys, such as the presence of a complete mure, an interrupted mesolophule, a metaloph that is connected to both the metaconule and posteroloph, a labial wall that closes the mesoflexus, and a mesostyle. Furthermore the first lower molar of “Phiomys” cf. lavocati has both a well-developed postcingulid and a lingual wall, and the posterior arm of the protoconid protrudes from the middle of the ectolophid, rather than from the protoconid as in Birkamys.

The diminutive species Phiocricetomys minutus, from Fayum Quarry I, differs from Birkamys in exhibiting strong distal reduction of the cheek teeth (i.e., loss of M3), having relatively rounded cusps, no metalophulid II, and an anterior end of the masseteric fossa that extends as far forward as the anterior end of the premolars. The mental foramen is placed beneath this ridge in Phiocricetomys, whereas in Birkamys the anterior end of the masseteric fossa lies just below the mental foramen. Phiocricetomys also has a cingulid that courses all the way around the labial surface of M1, and a strong anterocingulid that bears an anteroconid and some small accessory cusps. On the dP4 of Phiocricetomys, the metaconid and its posterior arm are more mesially positioned with respect to the protoconid, the anteroconid is well-developed, the metalophulid I is interrupted labially, the hypolophid is absent, the hypoconulid is well-developed and relatively large, with a posterior cingulid, and the posterolophid is very weak. The M2 of Phiocricetomys has a reduced entoconid, and a protoconid that is concave mesially and convex distally. The hypolophid is absent and the posterolophid is incipient, with no hypoconulid.

Contemporaneous Acritophiomys bowni (Sallam, Seiffert & Simons, 2012) is the same size as Mubhammys vadumensis, but it differs in replacing its deciduous premolars and in having well-developed mesolophules, double connections of the metaloph, complete mures, and relatively long lophs on the upper molars, while the lower molars bear well-developed posterior arms of the protoconids that approximate the lingual margins of the crowns. Moreover, the dP4 of A. bowni differs than that of M. vadumensis in having a well-developed anterocingulid, a complete posterior arm of the protoconid and a well-developed mesolophid, mesostylid, and ectostylid.

Birkamys and Mubhammys differ from the primitive hystricognaths Protophiomys and Waslamys from the earliest late Eocene (∼37 Ma) of the Fayum area (Sallam et al., 2009) in retaining deciduous premolars, having no mesolophid or mesoconid on dP4, in having lingually open mesoflexids on the lower molars, and, on the upper molars, lacking endolophs, mesolophules, and labial walls; Birkamys and Mubhammys also have M2 metalophs that are oriented distally, meeting the posterolophs.

In addition to its small size, Birkamys korai shares some dental features with Kahawamys mbeyaensis from the late Oligocene of the Rukwa Rift Basin in Tanzania (Stevens et al., 2009), such as a more centrally positioned ectolophid, a relatively mesial position of the entoconid with respect to the hypoconid, and a crest extending distally from the dP4 anteroconid to connect with the protoconid. The lower molars of Birkamys differ from those of Kahawamys in lacking an anterior arm of the entoconid, having relatively short and lingually open metaflexids, relatively distinct posterior arms of the protoconids that increase in length distally, anterocingulids on all molars, and a dP4 that is relatively wide, with a taller trigonid.

The extant cane rat Thryonomys resembles Birkamys in retaining deciduous premolars thoughout its life, but in addition to being much larger in size, differs from Birkamys and Mubhammys in having relatively short and hypsodont crowns, no anterocingulids or posterior arms of the protoconids on lower teeth, and in having a complete metalophulid I and a mesolophid in dP4. Furthermore, the M2 of Thryonomys differs from that of Birkamys and Mubhammys in having a complete mure and in lacking a metaconule. The M3 of Thryonomys differs from that of Birkamys in lacking a neo-endoloph, a metaloph and a well-developed metaconule.

Phylogenetic analysis

Parsimony analysis

Parsimony analysis in PAUP 4.0b10 with all characters equally weighted returned 10 equally parsimonious trees of length 907, the strict consensus of which is shown in Fig. 10A. In all trees, Birkamys and Mubhammys are nested deep within Phiomorpha as the sister clade of extant Thryonomys, with the sister group of the Birkamys-Mubhammys-Thryonomys clade containing early Oligocene “Paraphiomys” simonsi and Miocene Paraphiomys and Paraulacodus. With the placement of Canaanimys (late middle Eocene of Peru) taken to indicate the divergence of Caviomorpha from Phiomorpha, there is a pectinate sequence of basal phiomorphs that includes “Phiomys” hammudai and Turkanamys as its most basal members, followed sequentially by Acritophiomys, Prepomonomys, Phiomys, Neophiomys, Metaphiomys, and Diamantomys. Birkamys and Mubhammys are placed as sister taxa with strong support (bootstrap support (BS) = 82). Among derived hystricognaths, the only other clade that was supported by BS >50 was Canaanimys + Gaudeamus (late Eocene and early Oligocene of Africa) (BS = 62), Gaudeamus (BS = 99), and a Gaudeamus clade that excludes late Eocene Gaudeamus aslius (BS = 83). Elsewhere in the tree, the oldest African hystricognath, “Protophiomys” tunisiensis, is placed as the sister species of the phiocricetomyines Talahphiomys, “Phiomys” lavocati, and Phiocricetomys. The next-oldest species from northern Africa, from Locality BQ-2 (Protophiomys aegyptensis and Waslamys) and Bir el-Ater (Protophiomys algeriensis) are placed outside of the Phiomorpha-Caviomorpha clade, with P. algeriensis being the sister taxon of derived Asian “baluchimyines,” Waslamys forming a clade with “Protophiomys” durattalahensis, and Protophiomys aegyptensis intervening along the phiomorph-caviomorph stem between the divergences of the “P.” tunisiensis-phiocricetomyine clade and the Waslamys-“P.” durattalahensis clade.

Figure 10 Parsimony analysis of living and extinct hystricognathous rodents, based on 118 morphological characters, largely from the dentition, 77 of which were treated as ordered.

(A) Strict consensus of 10 MPTs recovered by parsimony analysis with all characters equally weighted. Tree length (TL) = 907, consistency index excluding uninformative characters (CI) = 0.2892, retention index (RI) = 0.5821, rescaled consistency index (RCI) = 0.1720. (B) Adams consensus of 119 MPTs recovered by parsimony analysis with transitions between “fixed” and polymorphic states weighted as a half-step; dashed branches break down in the strict consensus. TL = 595, CI = 0.3109, RI = 0.5769, RCI = 0.1847. (C) Maximum agreement subtree of all 119 trees summarized in (B). Taxon names are colored according to their geographical location. Bootstrap support values, based on 1,000 pseudoreplicates, are found above or below branches that are supported at a level >50%.

Parsimony analysis of the same data set with transitions between “fixed” and polymorphic states scaled to be equal to a half-step, rather than a full step (so that transitions between “fixed” states are equal to a single step) recovered 119 equally parsimonious trees of length 595. The strict consensus of these trees was largely unresolved (Fig. 10B), with only a Confiniummys-Ottomania clade, a Gaudeamus clade, a Bugtimys-Hodsahibia clade, a phiocricetomyine clade, and a Birkamys-Mubhammys clade being retained within Hystricognathi in the strict consensus. Notably, the sister-group relationship of Birkamys, Mubhammys, and extant Thyronomys recovered when all characters are equally weighted was not consistently recovered under the alternative weighting scheme; a maximum agreement subtree (Fig. 10C) excluded Thryonomys entirely and placed Birkamys and Mubhammys as the sister group of early Oligocene “Paraphiomys” simonsi and Miocene Paraphiomys. The maximum agreement subtree is also consistent with the strict consensus derived from parsimony analysis with all characters equally weighted (Fig. 10A).

Figure 11 “Allcompat” consensus (majority-rule plus compatible groups) of 50,000 post-burn-in trees retained by Bayesian analysis of the 118-character matrix in MrBayes 3.2.5.

Taxon names are colored according to their continental geographic location. Numbers above or below branches represent posterior probabilities (×100).

Bayesian phylogenetic analysis

As in the parsimony analyses, the “allcompat” (majority-rule plus compatible groups) consensus derived from the Bayesian analysis (Fig. 11) recovered a Birkamys-Mubhammys clade, with moderate support (posterior probability (PP) = 0.68), but this clade was not deeply nested within Phiomorpha, instead being the sister group of a well-supported (PP = 0.91) clade containing Oligocene (Metaphiomys, “Paraphiomys” simonsi) and Miocene (Diamantomys, Paraphiomys, Paraulacodus) species, as well as a Thryonomys-Prepomonomys clade. This result is completely consistent with the maximum agreement subtree derived from the parsimony analysis in which transitions between “fixed” and polymorphic states were scaled to a half-step. Also as in the parsimony analysis, the most basal phiomorphs are “Phiomys” hammudai and Turkanamys, followed by the sequentially more crownward genera Acritophiomys, Phiomys, and Neophiomys. The arrangement of species along the pectinate stem at the base of Phiomorpha in the allcompat tree is supported by PPs in the range of 0.49–0.56. Canaanimys and Gaudeamus form a well-supported clade (PP = 0.99), and “P.” tunisiensis is again placed as the sister taxon of phiocricetomyines, but with very weak support (PP = 0.36). There is strong support (PP = 0.91) for a clade containing African species, Canaanimys, and advanced “baluchimyines” to the exclusion of basal “baluchimyines” (Baluchimys ganeshaper, Baluchimys krabiense, Confinniumys, Lindsaya, Ottomania), but relationships among the basal members of the clade (Protophiomys, Waslamys) in the allcompat tree are only weakly supported (PPs = 0.25–0.5).

Tip-dating analysis with the fossilized birth–death prior

The “allcompat” consensus summarizing the 50,000 post-burn-in trees from the tip-dating analysis of the 118-character matrix with broad uniform priors on tip ages (i.e., analysis TD1, see Fig. S2) provided tip estimates that were averaged on a locality-by-locality basis for analysis TD2. These averages allowed for the ordering of localities from oldest to youngest as follows (Fig. 12, see Table 2 for 95% HPD): Subathu “Zone VIII” (India), 45.8 Ma; Khaychin II-III-IV (Mongolia), 43.5 Ma; Rencun Member (China), 41.0 Ma; Djebel el Kébar (Tunisia), 39.2 Ma; Birket Qarun Locality 2 (Egypt), 37.3 Ma; Bir el-Ater (Nementcha, Algeria), 37.3 Ma; Dur at-Talah DT1 (Libya), 36.2 Ma; Dur at-Talah DT2 (Libya), 35.9 Ma; Fayum Locality 41 (Egypt), 34.4 Ma; Fayum Quarries A and B (Egypt), 33.8 Ma; Krabi Bang Mark Pit (Thailand), 33.5 Ma; Lokone (Kenya), 31.9 Ma; Hsanda Gol (Mongolia), 32.0 Ma; Y-GSP 417 (Pakistan), 31.6 Ma; Süngülü (Turkey), 32.1 Ma; Fayum Quarry E (Egypt), 32.0 Ma; Fayum Quarry G (Egypt), 30.8 Ma; Paali Nala C2 (Pakistan), 29.9 Ma; Fayum Quarries I and M (Egypt), 29.6 Ma; and Silica North (Namibia), 25.7 Ma.

Figure 12 Uniform age priors for localities from which species in this analysis were sampled, with the sole or mean estimates provided by the tip-dating analysis (as output in the MrBayes “*.vstat” file).

Geological timescale was created using the program TSCreator 6.4 (http://www.tscreator.org).

Table 2 Estimated mean ages and upper and lower HPD intervals for species with relatively broad uniform age priors, derived from the tip-dating analysis 1 (TD1) with a fossilized birth-death prior.

Taxon	Locality	Mean	L 95% HPD	U 95% HPD	
Baluchimys ganeshaper	Y-GSP 417	31.66	28.11	33.90	
Baluchimys krabiense	Krabi, Bang Mark Pit	33.54	31.59	35.00	
Bugtimys zafarullahi	Paali Nala C2/Y-GSP 417(?)	30.21	25.64	33.90	
Confiniummys sidiki	Süngülü	32.76	28.68	37.20	
Hodsahibia gracilis	Paali Nala C2	29.64	24.93	33.90	
Lindsaya derabugtiensis	Y-GSP 417	31.72	28.16	33.90	
Lophibaluchia pilbeami	Y-GSP 417	31.34	27.09	33.90	
Ottomania proavita	Süngülü	31.40	28.10	35.58	
“Phiomys” hammudai	Dur at-Talah DT1	35.56	32.31	38.56	
Prepomonomys bogenfelsi	Silica North	25.71	20.00	30.41	
Protophiomys algeriensis	Bir el-Ater	37.27	33.90	40.45	
“Protophiomys” durattalahensis	Dur at-Talah DT1	37.60	35.49	39.60	
“Protophiomys” aff. durattalahensis	Dur at-Talah DT2	38.22	35.84	39.60	
“Protophiomys” tunisiensis	Djebel el Kébar	39.17	37.70	41.18	
Talahphiomys lavocati	Dur at-Talah DT2	33.65	31.00	36.60	
Talahphiomys libycus	Dur at-Talah DT1	35.31	31.20	38.87	
Turkanamys hexalophus	Lokone	31.88	28.30	33.90	

The resulting “allcompat” tree from analysis TD2, with tips fixed to the mean dates above (Fig. 13; see Fig. S3 and Dataset S5 for absolute median rates for each branch), is effectively the same as that from TD1 (Fig. S2; the sole difference being that Acritophiomys joins “Phiomys” hammudai, with very low probability (PP = 0.30)), but shows some notable differences from that based on the standard Bayesian analysis; few of the different placements are supported by high posterior probabilities, however. In contrast to the Bayesian allcompat consensus, the tip-dating allcompat (1) places Birkamys and Mubhammys in a slightly more basal position in phiomorph phylogeny, being the sister group of all younger phiomorphs aside from Phiomys; (2) shows much stronger support for the monophyly of progressively more nested phiomorph clades that include Turkanamys, “Phiomys” hammudai, and Acritophiomys (TD1 PP = 0.90, TD2 PP = 0.92, standard Bayesian PP = 0.55), Phiomys (TD1 PP = 0.90, TD2 PP = 0.91, standard Bayesian PP = 0.49), and Birkamys-Mubhammys (TD1 PP = 0.94, TD2 PP = 0.96, standard Bayesian PP = 0.50); (3) the sole undoubted caviomorph in the matrix (Canaanimys) in a particularly basal position, and with no special relationship to Gaudeamus, in strong contrast to the Bayesian analysis that supported a Canaanimys-Gaudeamus clade with a posterior probability of 0.99; (4) Tsaganomys and “basal baluchimyines” form a well-supported clade (TD1 PP = 0.87, TD2 PP = 0.92) rather than a paraphyletic stem with respect to African and South American hystricognaths (as found in both the standard Bayesian and parsimony analyses). Despite the differences in overall branching sequence between the tip-dating allcompat consensus and the standard Bayesian allcompat consensus, in the former the only higher-level clades that are supported by posterior probabilities >0.5 are Phiocricetomyinae (TD1 PP = 0.93, TD2 PP = 0.94), a clade of “advanced” baluchimyines (TD1 PP = 0.94, TD2 PP = 0.98), Gaudeamuridae (TD1 and TD2 PP = 1.0), and the clade including “phiomyids” and derived phiomorphs (TD1 PP = 0.90, TD2 PP = 0.92). As such, the interrelationships among those clades, Canaanimys, Waslamys and the various species assigned to the genus Protophiomys are not well-resolved, though support for a “Protophiomys” tunisiensis-Phiocricetomyinae clade increases to PP = 0.52 in analysis TD2.

Figure 13 “Allcompat” consensus (majority-rule plus compatible groups) of 50,000 post-burn-in trees retained by tip-dating analysis (“TD2”) of the 118-character matrix in MrBayes 3.2.5 with locality ages fixed based on sole or mean estimates calculated by the “TD1” analysis.

Branch lengths are colored according to rates of morphological evolution (calculated by multiplying the median rate for each branch, by the median clock rate for the entire tree), with the adjacent heat map showing the range of variation in the dataset. Taxon names are colored according to their continental geographic location. Numbers above or below branches represent posterior probabilities (×100).

Evolutionary rates (calculated for each branch as number of changes per site per Ma by multiplying the median rate for the branch in the allcompat consensus by the median estimate for the base clockrate that is output in the MrBayes *.pstat file; see Beck & Lee, 2014) are fairly consistent across most nodes in the tree, with the most striking accelerations being along the branches leading to Canaanimys and to the Birkamys-Mubhammys clade. The split between Caviomorpha and Phiomorpha (the latter in this case including basal baluchimyines and Tsaganomys) is estimated to have occurred at either 43.5 Ma (TD1) or 43.2 (TD2), and the largely African clade containing phiocricetomyines, gaudeamurids, and undoubted phiomorphs is estimated to have appeared at either 41.8 Ma (TD1) or 41.5 Ma (TD2). The advanced phiomorph clade that includes all species showing no evidence for P4∕4 eruption (i.e., the clade including Birkamys and Mubhammys but excluding Phiomys) is estimated to have appeared at 35.1 Ma in both TD1 and TD2.

Evolution of M1 size among early hystricognaths

With ln M1 area treated as a continuous variable evolving on the pruned tip-dating “allcompat” tree from TD2 (Fig. S4), the directional and random walk models (both with a lambda scaling parameter) returned similar log likelihoods, with little basis for preferring one model over the other. In Fig. 14, we present the results of runs based on both models, with mean estimates for ancestral nodes from each model represented by single points and the intervening space infilled to reflect differences in the mean estimates (see Dataset S6 for means and upper and lower 95% HPD for each node reconstructed).

Figure 14 Evolution of lower first molar area (on a natural log scale) (x-axis) through time (y-axis), based on ancestral values calculated in BayesTraits, using both directional and random walk models, each with a lambda scaling parameter.

Ancestral values from each model are represented by points, and the intervening space at each node is infilled to reflect uncertainty between the mean estimates.

While the two models leave considerable uncertainty about the mean M1 area estimate along the stem leading to the African hystricognath radiation, the estimate for that group’s common ancestor is well-constrained, with the random walk model recovering a mean estimate of 2.86 mm2 and the directional model a mean estimate of 2.69 mm2—i.e., about the size of Protophiomys aegyptensis from Locality BQ-2, and a little bit larger than the oldest Afro-Arabian hystricognath, “Protophiomys” tunisiensis (Marivaux et al., 2014). From this point of origin for African hystricognaths, there is an immediate size-related divergence between phiocricetomyines (here including “Protophiomys” tunisiensis) and the lineage leading to derived phiomorphs. Phiocricetomyines decrease in size at a fairly constant rate through time, finally terminating with the diminutive early Oligocene species Phiocricetomys minutus. There is little change leading to the initial divergence of Protophiomys sensu stricto, with slight increases in M1 area along the lineages leading to gaudeamurids, “Phiomys” hammudai, and Turkanamys, but in the early Priabonian a dwarfing event is implied, paralleling the trend seen in phiocricetomyines, along a trajectory that ultimately leads to the tiny species Phiomys andrewsi.

There is a reconstructed reversal of that trend in the Priabonian, close to the point of origin of the Birkamys-Mubhammys clade and the reconstructed acquisition of P4∕4 suppression, followed by gradual increases in size through the early Oligocene with the evolution of more derived and deeply nested phiomorphs (e.g., Metaphiomys). Given the trends reconstructed here, the small size of Neophiomys is, like Phiomys, also reconstructed as having been due to dwarfing, rather than retention of ancestral small size. Most remarkable in these size trends is the rapid divergence of Birkamys and Mubhammys from a common ancestor into dramatically different size categories, along trajectories that are almost horizontal on the size change versus time plot, showing that major change in size has occurred over only a very short period of time (note, though, that the implied change along the Birkamys branch appears somewhat exaggerated due to the use of logarithmically transformed variables). The upper and lower 95% HPDs for most ancestral estimates are quite broad (see Dataset S6), but the 95% HPD for the common ancestor of all Afro-Arabian hystricognaths (random walk model, 0.3205–1.8077; directional model, 0.2243–1.7649) nevertheless clearly excludes values in the range of Birkamys korai.

Discussion

Origin and evolution of the Birkamys-Mubhammys clade

The placement of Birkamys and Mubhammys as the exclusive sister taxa of extant Thryonomys in the maximum parsimony analysis in which all characters were equally weighted must be considered highly unlikely given the numerous extensive ghost lineages that the topology implies throughout phiomorph phylogeny (Fig. 15). When tips are scaled to the age estimates provided by the TD1 tip-dating analysis, and internodes are arbitrarily separated by 1 Ma, the strict consensus derived from the parsimony analysis with all characters equally weighted requires a total of 422.7 Ma along all branches (Fig. 15A, calculated in Mesquite 2.75 (“sum of branch lengths”); Maddison & Maddison, 2011), while the standard Bayesian allcompat tree, which places Birkamys and Mubhammys much more basally in phiomorph phylogeny, provides a significant reduction in overall time required (Fig. 15B, 357.4 Ma). However the placement of Thryonomys as the sister taxon of a Birkamys-Mubhammys clade in the unweighted parsimony analysis is not well-supported, and is sensitive to weighting of ordered characters; the maximum agreement subtree calculated from all trees recovered when transitions between “fixed” and polymorphic states are scaled to be a half-step rather than a full step does not include a Birkamys-Mubhammys-Thryonomys clade, and is entirely consistent with the Bayesian allcompat tree. When compared to the standard Bayesian allcompat tree, the tip-dating allcompat tree from TD2 requires only 45% of the total time accumulated across all branches (Fig. 15D, 159.4 Ma total). This tip-dating tree has several zones with very rapid divergences and short internodes, but even if branch lengths of terminal taxa and internodes are set to 0 in the parsimony tree (Fig. 15C), it still requires 125% more time (198.7 Ma versus 159.4 Ma) than the TD2 tip-dating tree.

Figure 15 “Sum of branch lengths” calculation across time-scaled topologies.

Tip-dating topology (D) compared to time-scaled strict consensus derived from parsimony analysis with all characters equally weighted (A) and Bayesian allcompat (B) topologies (species durations and internodes are arbitrarily scaled to 1 Ma) and (C) minimum (0) length parsimony topology.

Taken together, the phylogenetic and morphometric analyses presented here suggest that Birkamys and Mubhammys are members of a previously unrecorded late Eocene African lineage of early phiomorph rodents that diverged dramatically in size following a rapid change in tooth morphology in the later Eocene. The tip-dating analyses recovered particularly high evolutionary rates along the stem leading to the Birkamys-Mubhammys clade (Fig. 12), but ancestral reconstructions of M1 area indicate only a slight size decrease along this branch; instead the most dramatic size change is the rapid dwarfing event that is reconstructed along the branch leading to Birkamys. This pattern suggests that a relatively rapid change in dental morphology (overall simplification, involving the loss or reduction of transverse crests) might have facilitated, or driven, Birkamys’ expansion into a new niche space that was either unoccupied during the latest Eocene in this particular part of Africa, or that overlapped with the niche spaces of phiocricetomyines (small members of which have been recovered at L-41; descriptions currently in preparation).

The dwarfing event implied for the Birkamys lineage also suggests that the suppression of P4∕4 development and eruption likely first occurred at a body size larger than that of the tiny species B. korai, perhaps in populations with individuals that were about the size of Protophiomys aegyptensis or Pr. algeriensis. Given the ancestral reconstructions depicted in Fig. 14, initial suppression, however, would have evolved following an earlier, less dramatic, late Eocene dwarfing event from somewhat larger ancestors. After the Eocene-Oligocene boundary, members of the clade that evolved suppression subsequently undergo a steady increase in M1 size. Wood (1968) suggested that in early phiomorphs the P4∕4 might have “lagged in the race to become molariform” (p. 84), perhaps due to the longer retention of dP4∕4 in basal stem phiomorphs, and that life-long retention of dP4∕4 would have been advantageous if selection favored individuals with increased capacity for grinding across the post-diastemal dentition. The dwarfing event reconstructed prior to the evolution of P4∕4 suppression opens up the possibility that dP4∕4 retention might have first evolved in populations that were somewhat neotenous relative to their ancestors—i.e., containing individuals whose growth trajectories (including the timing of dental eruption) had been truncated, so that replacement of dP4∕4 ultimately never occurred. Only after P4∕4 suppression had been effectively “fixed” might there have then been selection for the increased hypsodonty of all unreplaced teeth in various Oligocene lineages. Given the presence of what might be yet another tiny phiomorph species at L-41 (cf. Birkamys, Fig. 6), however, it is also possible that the M1 size change scenario presented in Fig. 14 is overly simplistic; the hypothesis presented here will have to be tested with the recovery of additional species from late Eocene sites throughout Africa and Arabia.

Broader implications of tip-dating topologies for early hystricognath evolution

The parsimony, standard Bayesian, and tip-dating analyses all congruently supported a phiomorph clade that includes as its most basal members late Eocene Acritophiomys and “Phiomys” hammudai and Oligocene Turkanamys hexalophus. In the parsimony and standard Bayesian analyses, this clade was found to be the sister taxon of Caviomorpha — i.e., either a Gaudeamuridae + Caviomorpha clade (parsimony with all characters equally weighted) or a Gaudeamuridae + Caviomorpha +Waslamys + “Protophiomys” durattalahensis clade (standard Bayesian). In the tip-dating analysis, the caviomorph Canaanimys was not placed with gaudeamurids, and in fact was placed as the most basal of all hystricognaths, though with very weak support for its exclusion from more nested positions. One of the more remarkable topological rearrangements in the tip-dating analyses was the placement of primitive “baluchimyines” (Baluchimys, Confiniummys, Lindsaya, Ottomania) in a well-supported clade with early Oligocene Tsaganomys. This result is not entirely surprising, because the evidence for the paraphyly of the group with respect to derived hystricognaths is weak in the parsimony and standard Bayesian analyses (Figs. 10 and 11), and furthermore the paraphyly of the group requires extensive ghost lineages (Fig. 15) and presumably very low rates of evolution along terminal branches. However it is surprising that the monophyly of the group could be strongly supported (PP = 0.92 in the TD2 analysis) given these conditions. “Advanced” baluchimyines (Bugtimys, Hodsahibia, Lophibaluchia) are nested within the African hystricognath radiation across all analyses, implying an African origin for that clade and a dispersal to Asia, which is estimated to have occurred in the middle or late Eocene by the tip-dating analyses. However, while this topological result is strongly supported by the standard Bayesian analysis, it is not well-supported in either the parsimony or tip-dating analyses.

Telling time with rodent teeth: implications of tip estimates for the chronology of Paleogene hystricognath-bearing sites

In the absence of radioisotopically datable rocks, terrestrial mammal faunas from spatially and/or temporally isolated horizons can be extraordinarily difficult to date. The magnitude of the difficulty is proportional to the intensity of sampling of that temporal interval elsewhere on a landmass; if many other faunas of different ages bracket a fauna of interest, it is more likely that the same species will be recovered from multiple localities, and, in such cases, standard biochronology based on first/last appearances can be used to order localities, using for instance Appearance Event Ordination (Alroy, 1994). In the Paleogene of Afro-Arabia, sampling of the terrestrial mammal record has been so limited, and so patchy in space and time, that it is very rare for localities to show overlapping species (Seiffert, 2006; Seiffert, 2010; Coster et al., 2010). The taxonomy that workers choose to use can further obfuscate the situation; i.e., if newly discovered fossils are uncritically assigned to new species, there will be no species overlap for biochronological analysis.

These conditions have led to an unfortunate situation in which ordering of sites in the Paleogene of Africa is more often than not based on assumption-laden “stage of evolution” arguments that compare species from two localities and determine that one is older than the other because species A is “more primitive” than species B, or species B is of a more “advanced evolutionary stage” than species A. For instance, in attempting to determine the age of the Dur at-Talah localities in Libya that yielded some of the species included in our analysis, Jaeger et al. (2010a) argued that “Protophiomys is a primitive representative of the phiomyid African radiation and it is represented in Dur At-Talah by a slightly more derived species (Pr. durattalahensis) than that of Nementcha (Pr. algeriensis), thereby suggesting a younger age for the Dur At-Talah deposits” and that “the L41 rodent assemblage (Holroyd, 1994) contains more derived species than that of Dur at-Talah” (p. 211), leading them to argue that Dur at-Talah is also older than L-41. Sallam, Seiffert & Simons (2012, p. 297) argued that “Phiomys” hammudai from Dur at-Talah is perhaps slightly more primitive than Acritophiomys from L-41, but nevertheless is clearly more derived than the hystricognaths from BQ-2 (Waslamys attiai and Protophiomys aegyptensis)” in suggesting that Dur at-Talah is probably intermediate in age between L-41 and BQ-2.

Though such assessments might be based on compelling background information, the evidence and assumptions underlying the arguments are rarely explicit. On the broadest level, in the absence of a phylogenetic analysis, the comparison of the characters of species A with those of species B assumes that the two species are closely related and that the characters in question are homologous; it further assumes the evolutionary trajectory of the features, and that the presence of a presumed apomorphy in species B indicates that that feature has appeared more recently in time than the presumed plesiomorphic state in species A. These arguments thus also make assumptions about rates of evolution—they assume that a (presumed) plesiomorphic taxon species A is likely to be temporally older than a (presumed) apomorphic taxon species B because the reverse arrangement would imply relatively slow evolutionary rates in species A (i.e., stasis) and fast evolutionary rates in species B. It might even be assumed that the presence of a presumed apomorphy in species B implies that a certain amount of speciation must have occurred along the lineage leading to species B to account for that amount of change.

Bayesian tip-dating with the fossilized birth–death prior takes into account the important background information that must underlie these “stage of evolution” arguments—ages of related species, phylogenetic relationships among those species, rates of evolution, and patterns in speciation and fossilization—but in a much more explicit, objective, and replicable manner. Here we suggest that Bayesian tip-dating analysis with the fossilized birth–death prior is thus not only of use for dating internal nodes, but can also reasonably be “turned on its head” to provide age estimates for temporally poorly-constrained tips that have been assigned broad uniform priors on tip age—i.e., taking into account multiple biochronologically-relevant parameters to determine both phylogenetic position and the most likely point in time at which a morphological pattern would likely be present, given its broad uniform prior on age, its phylogenetic position, and the base clock rate of the tree.

One possible concern with this approach is that age estimates for species will simply fall near the middle of their uniform age priors. Our results clearly indicate that this is not the case. A particularly striking example is provided by Prepomonomys bogenfelsi, a species of contested age from the Sperrgebiet area of Namibia (Pickford et al., 2008). P. bogenfelsi and the other mammals from the Sperrgebiet Silica North locality were first described as Lutetian (early middle Eocene) in age (Pickford et al., 2008) and later as Bartonian (late middle Eocene) (Pickford et al., 2014), but multiple authors (Coster et al., 2010; Marivaux et al., 2014), including Pickford et al. (2008) themselves, have noted that the species from Silica North are similar to Miocene species from east Africa. Pickford et al. (2008) even placed some of the Silica North species in, or close to, otherwise exclusively Miocene phiomorph genera (Apodecter, cf. Bathyergoides). If the Silica North specimens are Bartonian in age as Pickford et al. (2014) suggest, their taxonomic identifications as Apodecter and cf. Bathyergoides would require ∼17–21-million-year-old temporal extensions for these genera far back into the Eocene, into time intervals when crown hystricognaths were (given a direct reading of the fossil record) only just starting to diversify. However, with our very conservative and broad 20–47 Ma uniform age prior on Prepomonomys, the TD1 analysis favored an age of 25.7 Ma (late Oligocene), i.e., far into the youngest part of the age prior, with a 95% HPD interval (20.0–30.41 Ma) that excludes most of the early Oligocene, and the Eocene entirely. Based solely on the fossils that have been described from Silica North and Silica South thus far (Pickford et al., 2008), and the strongly conflicting information provided by middle and late Eocene hystricognaths in northern Africa (Jaeger, Denys & Coiffait, 1985; Jaeger et al., 2010a; Marivaux et al., 2014; Sallam et al., 2009; Sallam, Seiffert & Simons, 2011; Sallam, Seiffert & Simons, 2012), a late Oligocene age would appear to us to make better sense of the rodent fauna from these localities, and could also explain why Silica North would have a relatively derived rodent fauna without any of the immigrant mammals that are thought to have arrived in Africa near the Oligocene-Miocene boundary (Rasmussen & Gutiérrez, 2009). The recently described mammals from the Eocliff and Eoridge sites (e.g., Pickford, 2015a; Pickford, 2015b), also in the Sperrgebiet area and also considered to be of Bartonian age by Pickford et al. (2014), are in our opinion also consistent with a late Oligocene age; the Eocliff tenrecoids (Pickford, 2015a) are morphologically intermediate between early Miocene species and those known from the late Eocene and Oligocene of Egypt (Seiffert & Simons, 2000; Seiffert et al., 2007; Seiffert, 2010), while the presence of a derived anthracotheriid artiodactyl (Pickford, 2015b)—an immigrant clade that has never been found at any of the earliest Priabonian Birket Qarun localities in the Fayum, and first appear in the Dir Abu Lifa Member of the stratigraphically younger Qasr el-Sagha Formation—strongly supports a maximum age of latest Priabonian for Eocliff, and more clearly an Oligocene age.

Similarly, Asian “baluchimyines” and Turkanamys have relatively old (early Oligocene) mean estimates within the broad (Oligocene-wide) uniform limits of their age priors. “Baluchimyine” species from the lower part (Bugti Member) of the Chitarwata Formation of Pakistan were initially thought to be early Miocene in age (Flynn, Jacobs & Cheema 1986), but there is now agreement that this unit is Oligocene, though estimates for its maximum age range from the early Oligocene into the late Oligocene (Lindsay et al., 2005; Métais et al., 2013). In the case of the five Chitarwata Formation “baluchimyine” species (of 12 known) sampled in this analysis, the resulting mean estimates (31.57 Ma for Y-GSP 417 and 29.9 Ma for Paali Nala C2) are consistent with the early Oligocene estimates of Welcomme et al. (2001) and Métais et al. (2013) based on biochronological interpretation of multiple vertebrate lineages. The mean age estimates for three species from Y-GSP 417 (as output in the MrBayes “.vstat” file) are tightly constrained at 31.7 Ma (28.1–33.9 Ma 95% HPD) for Baluchimys ganeshapher, 31.7 Ma (28.2–33.9 Ma 95% HPD) for Lindsaya derabugtiensis, and 31.3 Ma (27.1–33.9 Ma 95% HPD) for Lophibaluchia pilbeami, while Bugtimys zafarullahi and Hodsahibia gracilis from Paali Nala C2 were estimated to be 30.2 Ma (25.6–33.9 Ma 95% HPD) and 29.6 Ma (24.9–33.9 Ma 95% HPD), respectively. Lindsay et al. (2005) suggested that the baluchimyine-bearing base of the Chitarwata Formation is likely to be either ∼29.8 Ma or ∼25.8, and of these options our data support the former interpretation. Two other “baluchimyines” in the analysis, Confiniummys and Ottomania from Süngülü in Turkey, were thought to be close in age to the Eocene-Oligocene boundary (De Bruijn et al., 2003), but here were estimated to be about two million years younger (32.1 Ma), despite broad uniform priors extending from the end of the Oligocene (23 Ma) all the way back to the beginning of the late Eocene (37.8 Ma). Given the age estimates provided by the tip-dating analysis, Baluchimys krabiense from the Bang Mark Pit in Krabi, Thailand (Marivaux et al., 2000) would be the oldest “baluchimyine” from Asia at 33.5 Ma (31.6–35.0 Ma 95% HPD). Finally, the 31.9 Ma (28.3–33.9 Ma 95% HPD) estimate that Turkanamys hexalophus provides for the Lokone Hill sites in the Turkana Basin of Kenya is consistent with the broad late early to late Oligocene age suggested by Ducrocq et al. (2010). Seiffert (2012) suggested that the Lokone Hill sites were close in age to the boundary between the early and late Oligocene (∼28 Ma).

Finally, our analysis has implications for the age of the Dur at-Talah faunas DT1 and DT2, which have been described by Jaeger et al. (2010a) and Jaeger et al., (2010b) on “stage of evolution” grounds as being late middle Eocene (Bartonian) in age, but were instead considered to be late Eocene (Priabonian) by Sallam, Seiffert & Simons (2012), Antoine et al. (2012), and Marivaux et al. (2014). Both of these faunas present interesting problems in that they preserve remains of primitive species (“Protophiomys” durattalahensis at DT-1 and “Protophiomys” aff. durattalahensis at DT-2) that resemble Waslamys from BQ-2, combined with phiocricetomyines (Talahphiomys libycus (DT-1) and Talahphiomys lavocati (DT-2)) that have no relatives at BQ-2, and that more closely resemble species from the Oligocene levels of the Jebel Qatrani Formation. “Phiomys” hammudai from DT-1 shares derived features with Acritophiomys from the latest Eocene L-41 locality, as well as more derived phiomorphs from younger Fayum levels, and is not known from BQ-2. It is perhaps not surprising, then, that the tip estimates for the species from these localities are the most disparate of all in the analysis: the “Protophiomys” species are assigned mean estimates of 37.6 Ma (durattalahensis) and 38.2 Ma (aff. durattalahensis), which are in line with Jaeger et al.’s (2010a) late middle Eocene estimates, while the other Dur at-Talah species are estimated to be considerably younger: “Phiomys” hammudai is assigned a mean estimate of 35.6 Ma (1.7 Ma younger than the overall mean estimate for BQ-2), and Talahphiomys species are estimated to be even younger (35.3 Ma for T. libycus and 33.6 Ma for T. lavocati). Ultimately, the mean estimates based on all species are 36.2 Ma for DT-1 (3 species) and 35.9 Ma for DT-2 (2 species)—i.e., intermediate in age between BQ-2 and L-41, as was argued by Sallam, Seiffert & Simons (2012) and Seiffert (2012). The mean estimates for the ages of DT-1 and DT-2 do not require particularly fast or slow rates and are broadly consistent with adjacent branches (Fig. 13).

We would not argue that this method should be used in place of traditional biochronological methods that can be employed on landmasses with better sampling and sufficient species overlap, but it is certainly a more rigorous and repeatable approach than the data-free and assumption-laden “stage of evolution” arguments that have otherwise been applied to the ordering of terrestrial mammal sites in the very sparsely sampled Afro-Arabian Paleogene. We would expect this method to converge on increasingly robust age estimates as evidence is brought to bear from multiple biochronologically useful clades, and to be most useful when there are long and relatively well-dated reference sections available—for instance, in the case of Paleogene hystricognaths, the ability to integrate relatively tight age priors for species from the long Fayum succession undoubtedly helps to constrain several key parameters that in turn constrain estimates for poorly-constrained tips.

Incisive foramina of early phiomorphs

The most striking feature of the rostrum of Birkamys is the great enlargement of the apparently confluent incisive foramina, most clearly seen on the holotype specimen (CGM 66000, Fig. 3B) but also evident from the anteroposteriorly elongate, concave, and smooth lateral borders of the foramen on isolated maxillae [DPC 9276 (Fig. 4E) and DPC 15625 (Fig. 4B)]. The same pattern holds for the one maxilla of Mubhammys that preserves this area (Fig. 7C). An anteroposteriorly enlarged incisive foramen was also identified in Metaphiomys by Wood (1968, his Fig. 6) and referred to as an “anterior palatine fenestra”; of this, he noted (pp. 51–52) that “ventrally, the anterior palatine fenestrae are large and not sunk into a palatal depression as much as in Petromus or Thryonomys, although there is a shallow depression lateral to the fenestra (Fig. 6D), within which the fenestrae lie, which suggests the initial stages of a Petromus type of modification. The palatine fenestra is more rounded anteriorly than in the recent genera. No suggestion of an interpremaxillary foramen is present in any of the genera, although there is a paired foramen at the anterior end of the anterior palatine fenestra in Petromus and Thryonomys not seen in Metaphiomys. Posteriorly, there is a broad depression extending as far back as the anterior end of dP4 (Figs. 6A and 16), which seems to be identical to the deep fossa that contains the anterior palatine fenestrae in Petromus and Thryonomys.”

With the recovery of fairly complete, but crushed, crania of Acritophiomys and Gaudeamus from L-41 (Sallam, Seiffert & Simons, 2011; Sallam, Seiffert & Simons, 2012), it is now clear that enlarged incisive foramina are also present in those genera. The only known maxilla of Waslamys is not well-preserved, but appears to have a smooth margin anterior to the P4 that is similar to that of Birkamys (Fig. 16A). Among early Miocene taxa, similarly enlarged foramina are seen in Lavocatomys aequatorialis (Fig. 16F), Simonimys genovefae (Fig. 16H) and Paraphiomys stromeri. Diamantomys leuderitzi appears to have confluent foramina, but they are mediolaterally narrow and bordered by ventrally protruding flanges (Lavocat, 1973), perhaps correlated with the anteroposterior elongation of this region of the cranium. Kenyamys mariae also has more restricted foramina that are separated by a midline bony partition (Lavocat, 1973). In strong contrast, the incisive foramina of the Miocene bathyergoid Renefossor songhorensis are very small (Fig. 16C), and are either absent or tiny in Proheliophobius leakeyi (Fig. 16G). Among early Miocene bathyergoids, the foramina of Efeldomys loliae from Namibia (Mein & Pickford, 2008) appear to be the largest relative to tooth size (unless their apparent size is due to breakage), but are still much smaller than those of the L-41 species.

Figure 16 Comparison of the incisive foramina in late Eocene and early Miocene phiomorphs.

(A) CGM 66000, late Eocene Birkamys korai from Quarry L-41; (B) DPC 21311, late Eocene Acritophiomys bowni from Quarry L-41; (C) KNM-SO 710, early Miocene Renefossor songhorensis from Songhor, Kenya; (D) CGM 66006, late Eocene Gaudeamus aslius from Quarry L-41; (E) mirror-imaged maxilla of CGM 83690, late Eocene Waslamys attiai from Locality BQ-2; (F) KNM-SO 884, early Miocene Lavocatomys aequatorialis from Songhor, Kenya; (G) KNM-RU 2318, early Miocene Proheliophobius leakeyi from Rusinga Island, Kenya; (H) KNM-LG 834, early Miocene Simonimys genovefae from Legetet, Kenya.

From these observations the question arises as to whether the very enlarged foramina of Acritophiomys, Birkamys, Gaudeamus, and Mubhammys are primitive within Phiomorpha (or possibly even at some more inclusive level within Hystricognathi), or instead a synapomorphy of Thryonomyoidea (Petromus-Thryonomys) relative to Bathyergoidea, in which case Efeldomys and Renefossor retain the ancestral condition within Phiomorpha. The molecular divergence estimates of Patterson & Upham (2014) place the thryonomyoid-bathyergoid divergence at 36.3 Ma, but the divergence of gaudeamurids from other hystricognaths in the matrix is estimated by the tip-dating analyses to have occurred ∼39 Ma; assuming homology, this suggests an origin for such enlarged foramina well into the middle Eocene. There is no clear fossil record of the hystricid lineage before the Miocene, but, as in extant species, the late Miocene form “Hystrix” gansuensis has restricted foramina (Wang & Qui, 2002). Early Oligocene Tsaganomys, which is aligned with “baluchimyines” in the tip-dating analysis, has restricted foramina as well (Bryant & McKenna, 1995). Among early caviomorphs, Incamys bolivianus has enlarged foramina, while Branisamys luribayensis does not (Patterson & Wood, 1982 their Figs. 14 and 23). Unfortunately the origin of this distinctive feature cannot be adequately addressed without a matrix that samples more comprehensively from living and extinct ctenohystricans, but the recognition of the early ubiquity of this feature provides an interesting and easily identifiable new piece of evidence that will no doubt be of great importance for ongoing efforts to unravel early hystricognath phylogeny.

The broader question of the functional and behavioral implications of such variation will require much more research into the morphology of extant hystricognaths. The morphology seen in early phiomorphs might relate in some way to both the peculiar position of the vomeronasal organ of rodents, which opens anterior to the incisive foramina (Wöhrmann-Repenning, 1982; Giere, Freyer & Zeller, 1999), and the unique transformations of the rostral nasal skeleton and soft tissue structures of hystricognaths (Mess, 1999). For instance, Mess (1999) found that extant non-bathyergoid hystricognaths lacked fusion of the processus lateralis ventralis and the lamina tranversalis anterior, leading to a non-continuous rostral nasal floor, and that the nasal septum forms a ventrally protruding keel onto which facial musculature inserts; she suggested that this morphology could lead to increased mobility of the rostral nasal skeleton in these forms. In contrast, bathyergoids have a continuous rostral nasal floor and much smaller incisive foramina; of interest in this regard is the observation that among the bathyergoid phiomorphs, the naked mole-rat Heterocephalus has a very small vomeronasal organ, perhaps relating to eusociality (Smith et al., 2007).

Conclusions

The late Eocene genera described here, Birkamys and Mubhammys, further expand the morphological diversity observable in the terminal Eocene radiation of Afro-Arabian hystricognaths, and provide the first compelling evidence for a key phiomorph synapomorphy—suppression of P4∕4 eruption—having evolved by the latest Eocene. In their lower molar morphology, Birkamys and Mubhammys show considerable similarity to members of the genus Phiomys; this shared morphology likely would have characterized the late Eocene-aged last common ancestor of the phiomorph clade that contains Birkamys, Mubhammys, and Phiomys. The combined evidence from phylogenetic analysis and estimation of ancestral sizes of the first lower molar across early hystricognaths imply that Birkamys and Mubhammys underwent exceedingly rapid divergence in tooth size in the latest Eocene; the reasons for such a dramatic change are unclear. Cranial evidence from Birkamys, Mubhammys, and several Fayum and early Miocene has unexpectedly revealed that greatly enlarged incisive foramina are likely to be an ancient feature of phiomorph rodents, and possibly synapomorphic at an even more inclusive level within Hystricognathi.

Supplemental Information

Dataset S1 Character-taxon matrix for parsimony analysis of 42 hystricognath taxa, 118 characters

Click here for additional data file.

Dataset S2 Character-taxon matrix for Bayesian phylogenetic analysis of 42 hystricognath taxa, 118 characters

Click here for additional data file.

Dataset S3 Character-taxon matrix for Bayesian tip-dating analysis of 42 hystricognath taxa, 118 characters

Click here for additional data file.

Appendix S1 Justification for priors set on ages of included taxa in tip-dating analysis 1 (“TD1”)

Click here for additional data file.

Dataset S4 Data set for BayesTraits analysis of M1 size (natural logarithm of M1 area in mm2)

Click here for additional data file.

Figure S1 Deciduous versus permanent fourth premolars in early hystricognaths

Metaphiomys beadnelli, dP4 (DPC 2431, reversed) and dP4 (DPC 6047, reversed); Mubhammys vadumensis, dP4 (DPC 14324) and dP4 (DPC 13220); Birkamys korai, dP4 (DPC 17457, reversed) and dP4 (DPC 22737, reversed); Gaudeamus aslius, dP4 (DPC 15748, reversed), P4 (CGM 66006, reversed), P4 (DPC 17677, reversed), and dP4 (DPC 16920); Acritophiomys bowni, dP4 (DPC 12662, reversed), P4 (CGM 83705, reversed), P4 (DPC 14168), and dP4 (DPC 20288, reversed); Waslamys attiai, dP4 (DPC 21365E), P4 (DPC 23305G, reversed), P4 (DPC 21371N), and dP4 (DPC 21293P).

Click here for additional data file.

Figure S2 “Allcompat” consensus (majority-rule plus compatible groups) of 50,000 post-burn-in trees retained by tip-dating analysis (“TD1”, wide priors on tip ages) of the 118-character matrix in MrBayes 3.2.5

Click here for additional data file.

Figure S3 “Allcompat” topology derived from TD2 tip-dating analysis, with branches numbered; absolute median rate for each branch is provided in Supporting Dataset S2

Click here for additional data file.

Dataset S5 Absolute median rates for each branch numbered in Supporting Fig. 2

Click here for additional data file.

Figure S4 Tree used for Bayesian reconstruction of ln M1 area throughout basal hystricognath phylogeny

Means and 95% upper and lower HPD are provided for each numbered node in Supporting Dataset S3.

Click here for additional data file.

Dataset S6 Means and 95% upper and lower HPD for ancestral estimates of ln M1 area, for nodes labelled in Supporting Fig. 3

Click here for additional data file.

We thank G Gunnell, P Chatrath, and C Riddle for access to the fossil collection at the Division of Fossil Primates, Duke Lemur Center, FK Manthi for access to Miocene phiomorphs at the National Museums of Kenya, and M Abdel Ghany and AH Sileem for access to specimens at the CGM. J Thostenson (Duke) and PM O’Connor (Ohio University) provided access to micro-CT scanning facilities. ME Frenkel helped with SEM facilities housed at the American Museum of Natural History. We thank L Marivaux and ME Pérez for their helpful reviews of the manuscript, and A Hammond, J Herrera, and N Stevens for helpful discussion. We thank the staff of the Egyptian Mineral Resources Authority and the Egyptian Geological Survey for facilitating our work in the Fayum area. This is Duke Lemur Center publication # 1313.

Institutional Abbreviations

CGM Egyptian Geological Museum, Egypt

DPC Duke Lemur Center Division of Fossil Primates

MUVP Mansoura University Vertebrate Paleontology Center

Additional Information and Declarations

Competing Interests

Author Contributions

Data Availability

New Species Registration

1 We exclude from our discussion two molars from the Silica North and Silica South localities in the Sperrgebiet area of Namibia that were assigned by Pickford et al. (2008) to the otherwise late Eocene genus Protophiomys. Pickford et al. (2014) argue that Silica North and Silica South are Bartonian (∼38–41.3 Ma; Gradstein et al., 2012) in age, but the evidence that we present later in this contribution suggests a much younger age (late Oligocene). Marivaux et al. (2014, p. 14) suggested an even younger (Miocene) age for these deposits: “…because of the abundance of associated rodent species that are clearly of Miocene affinities, the presence of Protophiomys in Sperrgebiet, and the middle Eocene age of the Silica rodent-bearing localities are questionable. These Silica localities are most likely Miocene in age, and as such the alleged “Protophiomys”-like teeth are certainly referable to another genus.” In the absence of much more complete material, we view the great age proposed for the Sperrgebiet Protophiomys specimens with similar skepticism.

The authors declare there are no competing interests.

Hesham M. Sallam conceived and designed the experiments, analyzed the data, wrote the paper, prepared figures and/or tables, reviewed drafts of the paper.

Erik R. Seiffert conceived and designed the experiments, performed the experiments, analyzed the data, wrote the paper, prepared figures and/or tables, reviewed drafts of the paper.

The following information was supplied regarding data availability:

http://morphosource.org/index.php/Detail/ProjectDetail/Show/project˙id/184.

The following information was supplied regarding the registration of a newly described species:

Publication LSID: urn:lsid:zoobank.org:pub:9DB0476B-E752-4EA1-8745-95 8C92E429C65B.

Birkamys, new genus: urn:lsid:zoobank.org:act:D8D841E7-55E1-4F3A-9B41-CE37F8B6F3A7

Birkamys korai, new species: urn:lsid:zoobank.org:act:CF09DD3A-E265-43FA-85B6-E9868F9B8364

Mubhammys, new genus: urn:lsid:zoobank.org:act:BAE3043A-8B71-441F-B0C3-9A0B7E6CC630

Mubhammys vadumensis, new species: urn:lsid:zoobank.org:act:1E6685EA-BAF3-4035-B62D-0D0C486A3D46.

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
