# Peer review of "New phiomorph rodents from the latest Eocene of Egypt, and the impact of Bayesian “clock”-based phylogenetic methods on estimates of basal hystricognath relationships and biochronology"

_PeerJ, doi:10.7717/peerj.1717_

## Round 0.1 · original submission · Minor Revisions

The reviews for your manuscript are very favorable. However, it is very important for you to address all of their comments and criticisms in your revised manuscript and in your rebuttal letter. In particular, please add a section in which you describe and defend your choice of Bayesian methods in your analysis (see comments from Reviewer #2). Please also note that Reviewer #1 added comments in an annotated file.

·

Basic reporting

No comments

Experimental design

-Original primary research within the scope of the journal;
-The investigation has been conducted rigourously;
-The MS reports critical new data, which shed new light on the early evolutionary history of hystricognathous rodents.
-The methods are abundantly and correctly detailed;

Validity of the findings

-the authors have performed a set of analyses, somewhat new, in employing the statistical assessments associated;
-All the data used for reaching the conclusions of the paper are available and posted as sup. info.;
-The discussion of the paper focus on the main questions addressed, and is based on the new data and the phylogenetic results obtained by the authors;

Additional comments

In this project of paper for “PeerJ”, Hesham Sallam and Erik Seiffert report new and very interesting fossil of hystricognathous rodents from a well-known mammal-bearing locality (L-41) in the Fayum depression of Egypt. This locality, dating from the latest Eocene, is particularly important since it is situated near the Eocene/Oligocene transition, a period which is not well-represented in Africa and that recorded a global cooling. In this paper, Sallam and Seiffert describe two new genera based on cranial, mandibular and dental materials. They propose a cladistic (via PAUP) assessment of the morphological evidence available (primarily dental), and use a set of new approaches (Bayesian and “Tip-Dating”) for replacing these new taxa within a phylogenetic context. They discuss the phylogenetic results as well as some macroevolutionary implications and character transformations.

The results are well-presented and include appropriate/informative figures. I would like to acknowledge the authors for the high quality of the fossil specimen illustrations (CTscans and SEM). The figures are abundant and necessary. The anatomical descriptions are detailed, but I have noticed several mistakes and imprecisions in the descriptions of the material that I have directly corrected (suggested to be corrected) on the PDF of the MS.

I have no fundamental disagreement with the interpretation, but I remain somewhat skeptical regarding the “deciduous” versus “permanent” premolar identification of the authors. This is a very critical point that the authors discuss. But, I am not sure that the fossils are abundant enough to provide decisive conclusions about the retention of the deciduous premolars. For me, the status of these mesialmost teeth on the toothrows is ambiguous. It is difficult for me to clearly identify the deciduous and the permanent premolars, notably regarding the lower specimens. I would suggest that the authors provide much more arguments for justifying their opinion on the identification of these teeth (deciduous versus permanent). Why the premolars are described as deciduous? What are the main characters that allow for such an attribution?

The manuscript is rather well-written but need a set of improvements and adjustments. I have addressed several questions and remarks through the MS. All these comments are directly written on the PDF MS that I join to this short report. The manuscript is somewhat long, but the length is justified for the topics covered with no pruning necessary.

Would you please find associated to this message the annotated PDF of the MS, as well as the Word document furnished in sup info (that I have pasted at the end of the long PDF-MS main text). I think that this work should be published after addressing the points underscored directly on the PDF MS. I hope that the numerous suggestions I propose concerning different aspects of the manuscript will serve to the authors for improving and adjusting their text.

There is no doubt that this paper will be of interest to the readers of “PeerJ”. Therefore, I recommend the publication of this article after revisions!

Laurent Marivaux

·

Basic reporting

No Comments.

Experimental design

No Comments.

Validity of the findings

The authors are working with paleontological data, for this reason they must provide a better justification of its choosing the probability approaches (Bayesian methods in this case) over parsimony method. Mainly because they make very important evolutionary inferences, post-analyzes.

Additional comments

I have reviewed the manuscript "New phiomorph rodents from the latest Eocene of Egypt, and the impact of Bayesian “clock”-based phylogenetic methods on estimates of basal hystricognath relationships and biochronology" and it is a really valuable and very interesting contribution by Hesham Sallam and Erik Seiffert, about two new phiomorph rodents including the phylogenetic and biochronological implications in the context of the Hystricognathi rodents.

The work includes the taxonomic analysis of two new genera and species of phiomorphs (Birkamys korai and Mubhammys vadumensis) and a very detailed description and comparison with other phiomorph rodents, including very good figures of the new specimens.

On the other hand, the authors analyze through different methodologies (i.e. parsimony, standard Bayesian, and tip-dating) the phylogenetic position of the two new species in the context of phiomorphs, they infer the age of early hystricognath of some localities that are not well limited (for example at biostratigraphical level), they analyze the evolution of the size of M1, and the incisive foramina in the early phiomorph rodents.

The literature cited, figures, tables, and supplementary information are good and necessary.

Although I recommend this work for publication, I consider that a major issue is the use of Bayesian methods in paleontological data sets.
The authors must provide a better justification of its choosing the probability approaches (Bayesian methods in this case) over parsimony method. Mainly because they make very important evolutionary inferences, post-analyzes.

The paper of Xu and Pol (2012: Xu, X. and Pol, D. 2012. Archaeopteryx, paravian phylogenetic analyses, and the use of probability-based methods for palaeontological datasets. Journal of Vertebrate Paleontology http://dx.doi.org/10.1080/14772019.2013.764357) explain clearly that the use of Bayesian methodologies are inadequate on paleontological datasets, providing many examples and justifications about it.

For example, Xu and Pol (2012: p.3-4) explain: "Probability-based methods arewidely used inmolecular systematics and they have been demonstrated to be superior to other methods under some specific circumstances (i.e. when characters evolve under a homogeneous Markov model) (Huelsenbeck&Hillis 1993; Huelsenbeck 1995; Swofford et al. 2001; Felsenstein 2004; Huelsenbeck et al. 2011). However, even in ideal cases in which the evolutionary model perfectly fits the assumed model of the probabilistic method, there are particular cases or biases that affect the efficiency of both maximum likelihood (Pol & Siddall 2001) and Bayesian analyses (Pickett & Randle 2005; Goloboff & Pol 2005). A relevant point here is that when the analysed data deviates from the homogeneity imposed by the homogeneous Markov model, probability-based methods that assume such models can be severely affected (e.g. leading to statistical inconsistency) (Chang 1996). In fact, parsimony has been shown to perform better in simulations where characters evolve heterogeneously (Kolaczkowski & Thornton 2004; Goloboff & Pol 2005; Simmons et al. 2006), which is a reasonable scenario for morphological character evolution."

Sallam and Seiffert explain (lines 854-856 ): "The placement of Birkamys and Mubhammys as the exclusive sister taxa of extant Thryonomys in the maximum parsimony analysis must be considered highly unlikely given the numerous extensive ghost lineages that the topology implies throughout phiomorph phylogeny".
I consider that the choise of a method over other should be the analysis and comparison of characters that support the different methodologies (In this sense, Xu and Pol (2012: 3) explain that: "...the recovery of a preferred topological result is irrelevant for recommending the use of a phylogenetic method.")

Sincerely,

María Encarnación Pérez

---

## Round 0.2 · accepted · Accept

Thank you for your revisions.

·

Basic reporting

No comments.

Experimental design

No comments.

Validity of the findings

No comments.

Additional comments

The manuscript "New phiomorph rodents from the latest Eocene of Egypt, and the impact of Bayesian “clock”-based phylogenetic methods on estimates of basal hystricognath relationships and biochronology" (by Hesham Sallam and Erik Seiffert) has returned for a second revision round.

The authors have accepted the suggestions of both reviewers improving and clarifying the manuscript (in respect the first version). They included better explaining about the identification of deciduous and permanent premolars in the early Hystricognathi and about metodologies, and they added also new images, figures, and supplementary information.

Personally, I think that it would have been very interesting include a discussion about the use of Bayesian methods with paleontological datasets in this contribution, but I understand that maybe would be too extensive for this manuscript.

I RECOMMEND this contribution by Hesham Sallam and Erik Seiffert for publish in PeerJ.

Sincerely,

María Encarnación Pérez